# A neural-mechanistic hybrid approach improving the predictive power of genome-scale metabolic models

Léon Faure[1], Bastien Mollet[2,3], Wolfram Liebermeister [4] & Jean-Loup Faulon [1,5] ✉

Constraint-based metabolic models have been used for decades to predict the phenotype of microorganisms in different environments. However, quantitative predictions are limited unless labor-intensive measurements of media uptake fluxes are performed. We show how hybrid neural-mechanistic models can serve as an architecture for machine learning providing a way to improve phenotype predictions. We illustrate our hybrid models with growth rate predictions of *Escherichia coli* and *Pseudomonas putida* grown in different media and with phenotype predictions of gene knocked-out *Escherichia coli* mutants. Our neural-mechanistic models systematically outperform constraint-based models and require training set sizes orders of magnitude smaller than classical machine learning methods. Our hybrid approach opens a doorway to enhancing constraint-based modeling: instead of constraining mechanistic models with additional experimental measurements, our hybrid models grasp the power of machine learning while fulfilling mechanistic constrains, thus saving time and resources in typical systems biology or biological engineering projects.

In this study, we present an approach that combines machine learning (ML) and mechanistic modeling (MM) to improve the performance of constraint-based modeling (CBM) on genome-scale metabolic models (GEMs). Our hybrid MM-ML models are applied to common tasks in systems biology and metabolic engineering, such as predicting qualitative and quantitative phenotypes of organisms grown in various media or subjected to gene knock-outs (KOs). Our approach leverages recent advances in ML, MM, and their integration, which we briefly review next.

The increasing amounts of data available for biological research bring the challenge of data integration with ML to accelerate the discovery process. The most compelling achievement within this grand challenge is protein folding, recently cracked by AlphaFold[1], which in the last CASP14 competition predicted structures with a precision similar to structures determined experimentally. Following

this foot step, one may wonder if in the future we will be able to use ML to accurately model whole-cell behaviors. The curse of dimensionality[2], i.e. the fact that fitting many parameters may require prohibitively large data sets, is perhaps the biggest hurdle that prevents using ML to build cell models. Obviously, cells are far more complex than single proteins and since the amount of data needed for ML training grows exponentially with the dimensionality[2], as of today, ML methods have not been used alone to model cellular dynamics at a genome scale.

For the past decades, MM methods have been developed to simulate whole-cell dynamics (cf. Thornburg et al.[3] for one of the latest models). These models encompass metabolism, signal transduction, as well as gene and RNA regulation and expression. Cellular dynamics being tremendously complex, MM methods are generally based on strong assumptions and oversimplifications. Ultimately, they suffer

[1]MICALIS Institute, INRAE, AgroParisTech, University of Paris-Saclay, 78350 Jouy-en-Josas, France. [2]Ecole Normale Supérieure of Lyon, 69342 Lyon, France. [3]UMR MIA, INRAE, AgroParisTech, University of Paris-Saclay, 91120 Palaiseau, France. [4]MaIAGE, INRAE, University of Paris-Saclay, 78350 Jouy-en-Josas, France. [5]Manchester Institute of Biotechnology, University of Manchester, Manchester M1 7DN, UK. ✉e-mail: Jean-loup.Faulon@inrae.fr

from a lack of capacities to make predictions beyond the assumptions and the data used to build them.

Flux balance analysis (FBA) is the main MM approach to study the relationship between nutrient uptake and the metabolic phenotype (i.e., the metabolic fluxes distribution) of a given organism, e.g., *E. coli*, with a model iteratively refined over the past 30 years or so[4]. FBA searches for a metabolic phenotype at steady state, i.e., a phenotype that is constant in time and in which all compounds are mass-balanced. Usually, such a steady state is assumed to be reached in the mid-exponential growth phase. The search for a steady state happens in the space of possible solutions that satisfies the constraints of the metabolic model, i.e., the mass-balance constraints according to the stoichiometric matrix as well as upper and lower bounds for each flux in the distribution. The steady state search is performed with an optimality principle, with one principal objective (usually the biomass production flux) and possibly secondary objectives (e.g., minimize the sum of fluxes in parsimonious FBA, or the flux of a metabolite of interest). As we shall see later and as discussed in O'Brien et al.[5], FBA suffers from making accurate quantitative phenotype predictions.

The MM and ML approaches are based on two seemingly opposed paradigms. While the former is aimed at understanding biological phenomena with physical and biochemical details, it has difficulties handling complex systems; the latter can accurately predict the outcomes of complex biological processes even without an understanding of the underlying mechanisms, but require large training sets. The pros of one are the cons of the other, suggesting the approaches should be coupled. In particular, MMs may be used to tackle the dimensionality curse of ML methods. For instance, one can use MMs to extend experimental datasets with in silico data, increasing the training set sizes for ML. However, with that strategy, if the model is inaccurate, ML will be trained on erroneous data. One can also embed MMs within the ML process, in this strategy, named hybrid-modeling, ML and MM are trained together and the model parameters can be estimated through training, increasing the model predictive capacities. To improve FBA phenotype predictions, ML approaches have been used to couple experimental data with FBA. Among published approaches, one can cite Plaimas et al.[6] where ML is used after FBA as a post-process to classify enzyme essentiality. Similarly, Schinn et al.[7] used ML as a post-process to predict amino acid concentrations. Freischem et al.[8] computed a mass flow graph running FBA on the *E. coli* model iML1515[9] and used it with a training set of measured growth rates on *E. coli* gene KO mutants. Several ML methods were then utilized in a post-process to classify genes as essential vs. non-essential. As reviewed by Sahu et al.[10], ML has also been used to preprocess data and extract features prior to running FBA. For instance, data obtained from several omics methods can be fed to FBA, after processing multi-omics data via ML[11–13].

In all these previous studies, and as discussed in Sahu et al.[10], the interplay between FBA and ML still shows a gap: some approaches use ML results as input for FBA, others use FBA results as input for ML, but none of them embed FBA into ML, as we do in this study with the artificial metabolic network (AMN) hybrid models.

The main issue with hybrid modeling is the difficulty of making MM amenable to training. Overcoming this difficulty, solutions have recently been proposed under different names in biology for signaling pathways and gene-regulatory networks (Knowledge Primed Neural Network[14], Biologically-Informed Neural Networks[15]) with recent solutions based on recurrent neural networks (RNNs)[16]. Hybrid models have also been developed in physics to solve partial differential equations, such as Physics Informed Neural Network[17] (PINN), available in open-source repositories like SciML.ai[18]. The goal of these emerging hybrid modeling solutions is to generate models that comply well with observations or experimental results via ML, but that also use mechanistic insights from MM. The advantages of hybrid models are two-fold: they can be used to parametrize MM methods through direct training and therefore increasing MM predictability, and they enable ML methods to overcome the dimensionality curse by being trained on smaller datasets because of the constraints brought by MM.

In the current paper we propose a MM-ML hybrid approach in which FBA is embedded within artificial neural networks (ANNs). Our approach bridges the gap between ML and FBA by computing steady-state metabolic phenotypes with different methods that can be embedded with ML. All these methods rely on custom loss functions surrogating the FBA constraints. By doing so, our AMNs are mechanistic models, determined by the stoichiometry and other FBA constraints, and also ML models, as they are used as a learning architecture.

We showcase our AMNs with a critical limitation of classical FBA that impede quantitative phenotype predictions, the conversion of medium composition to medium uptake fluxes[5]. Indeed, realistic and condition-dependent bounds on medium uptake fluxes are critical for growth rate and other fluxes computations, but there is no simple conversion from extracellular concentrations, i.e., the controlled experimental setting, to such bounds on uptake fluxes. With AMNs, a neural pre-processing layer aims to capture, effectively, all effects of transporter kinetics and resource allocation in a particular experimental setting, predicting the adequate input for a metabolic model to give the most accurate steady-state phenotype prediction possible. Consequently, AMNs provide a new paradigm for phenotype prediction: instead of relying on a constrained optimization principle performed for each condition (as in classical FBA), we use a learning procedure on a set of example flux distributions that attempts to generalize the best model for accurately predicting the metabolic phenotype of an organism in different conditions. As shown in the results section, the AMN pre-processing layer can also capture metabolic enzyme regulation and in particular predict the effect of gene KOs on phenotype.

## Results
### Overview of AMN hybrid models

When making predictions using FBA, one typically sets bounds for medium uptake fluxes, $\mathbf{V_{in}}$, to simulate environmental conditions for the GEM of an organism (Fig. 1a). Each condition is then solved independently from each other by a linear program (LP), usually making use of a Simplex solver. In most cases, one sets the LP's objective to maximize the biomass production rate (i.e., the growth rate), under the metabolic model constraints (i.e., flux boundary and stoichiometric constraints). FBA computes the resulting steady-state fluxes, $\mathbf{V_{out}}$, for all the reactions of the metabolic network, which we use later in our reference "FBA-simulated data", for the benchmarking of the hybrid models developed in this study. While FBA is computationally efficient and easy to use through libraries like Cobrapy[19], FBA cannot directly be embedded within ML methods, like neural networks, because gradients cannot be backpropagated through the Simplex solver.

To enable the development of hybrid models and gradient back-propagation, we developed three alternative MM methods (Wt-solver, LP-solver and QP-solver) that replace the Simplex solver while producing the same results (Fig. 1b). The three solvers, further described in the next subsection, take as input any initial flux vector that respect flux boundary constraints.

We next used the MM models as a component of AMN hybrid models that can directly learn from sets of flux distributions (Fig. 1c). These flux distributions used as learning references (i.e., training sets) are either produced through FBA simulations or acquired experimentally. The AMN model comprises a trainable neural layer followed by a mechanistic layer (composed of Wt-solver, LP-solver or QP-solver). The purpose of the neural layer is to compute an initial value, $\mathbf{V^0}$, for the flux distribution to limit the number of iterations of the mechanistic layer. The initial flux distribution is computed from medium uptake flux bounds, $\mathbf{V_{in}}$, when the training set has been

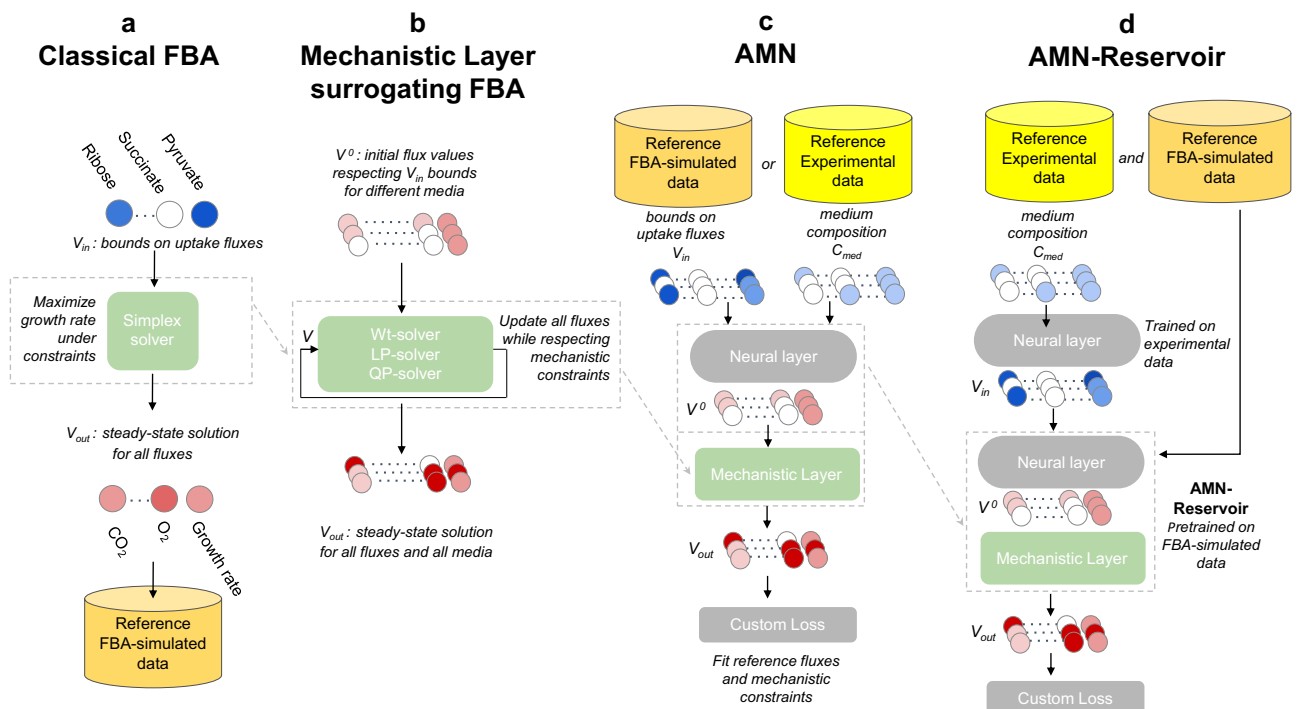

**Fig. 1 | Computing and learning frameworks for FBA, alternative mechanistic models, AMN, and AMN-Reservoir. a** Computing framework for classical FBA. The process is repeated for each medium, computing the corresponding steady state fluxes. Blue circles represent different bounds on metabolites uptake fluxes and each red circle represents a flux value at steady-state. **b** Computing framework for MM methods surrogating FBA. The methods can handle multiple growth media at once. Disregarding the solver (Wt, LP and QP), the MM layer takes as input an arbitrary initial flux vector, $V^0$, respecting uptake flux bounds for different media, and computes all steady-state fluxes values ($V_{out}$) through an iterative process. **c** Learning framework for AMN hybrid models. The input (for multiple growth media) can be either a set of bounds on uptake fluxes ($V_{in}$), when using simulation data (generated as in **a**), or a set of media compositions, $C_{med}$, when using

experimental data. The input is then passed to a trainable neural layer, predicting an initial vector, $V^0$, for the mechanistic layer (a MM method of **b**). In turn, the mechanistic layer computes the final output of the model, $V_{out}$. The training is based on a custom loss function (cf. "Methods") ensuring the reference fluxes are fitted (i.e., $V_{out}$ matches simulated or measured fluxes) and that the mechanistic constraints (on flux bounds and stoichiometry) are respected. **d** Learning framework for an AMN-Reservoir. The first step is to train an AMN on FBA-simulated data (as in **c**), after which parameters of this AMN are frozen. This AMN model, which purpose is to surrogate FBA, is named non-trainable AMN-Reservoir. In the second step, a neural layer is added prior to $V_{in}$ taking as input media compositions, $C_{med}$, and learning the relationship between the compositions and bounds on uptake fluxes.

generated through FBA simulations, or medium compositions, $C_{med}$, for experimental training sets. For all AMNs, the training of the neural layer is based on the error computation between the predicted fluxes, $V_{out}$, and the reference fluxes, as well as on the respect of mechanistic constraints. It is important to point out that AMNs attempt to learn a relationship between $V_{in}$ (or $C_{med}$) and the steady-state metabolic phenotype, generalizing this relationship for a set of conditions and not just only one as in FBA. In the upcoming subsections, Fig. 2 presents results for FBA-simulated training sets and Figs. 3 and 4 results for experimental training sets.

Finally, we developed a non-trainable AMN-Reservoir to showcase how the predictive power of classical FBA can be improved (Fig. 1d). This architecture is based on a two-step learning process with the specific goal of finding the best bounds on uptake fluxes for FBA, by extracting $V_{in}$ after training. Indeed, once the AMN has been trained on adequate FBA-simulated data, we can fix its parameters, resulting in a gradient backpropagation compatible reservoir that mimics FBA. The AMN reservoir can then be used to tackle the above-mentioned issue of unknown uptake fluxes: adding a pre-processing neural layer and training this layer with an experimental dataset, one can predict uptake fluxes from the media composition. Results of the pre-processing neural layer can directly be plugged into a classical FBA solver and the neural layer can be reused by any FBA user to improve the predictive power of metabolic models with an adequate experimental set-up. We showcase AMN-Reservoir results in Fig. 5 using experimental measurements acquired on *E. coli* and *P. putida*.

## Alternative mechanistic models to surrogate FBA

Let us first recall that the methods described in this subsection are mechanistic models (MMs) that replace the Simplex-solver used in FBA and allow for gradient backpropagation, but without any learning procedure performed. As far as medium uptake fluxes are concerned, we consider in the following two cases: (1) when exact bound values (EB) for medium uptake fluxes are provided, and (2) when only upper bound values (UB) for medium uptake fluxes are given.

Our first method (Wt-solver), inspired by previous work on signaling networks[16], recursively computes **M**, the vector of metabolite production fluxes, and **V**, the vector of all reaction fluxes (cf. "Wt-solver" in "Methods" and in Supplementary Information for further details). The vectors **M** and **V** are iteratively updated using matrices derived from the metabolic network stoichiometric matrix **S** and from a weight matrix, $W_r$, representing consensual flux branching ratios found in example flux distributions (i.e., reference FBA-simulated data or experimental measurements). Since the mass conservation law is the central rule when satisfying metabolic networks constraints, these ratios play a key role in the determination of the metabolic phenotype, i.e. the paths taken by metabolites in the organism. In this approach, we assume that the flux branching ratios remain similar between flux distributions with different bounds on different uptake fluxes. A simple toy model network is shown to demonstrate the functioning of the Wt-solver in Supplementary Fig. S1.

While the Wt-solver is simple to implement it suffers from a drawback. As discussed in Supplementary Information "AMN-Wt

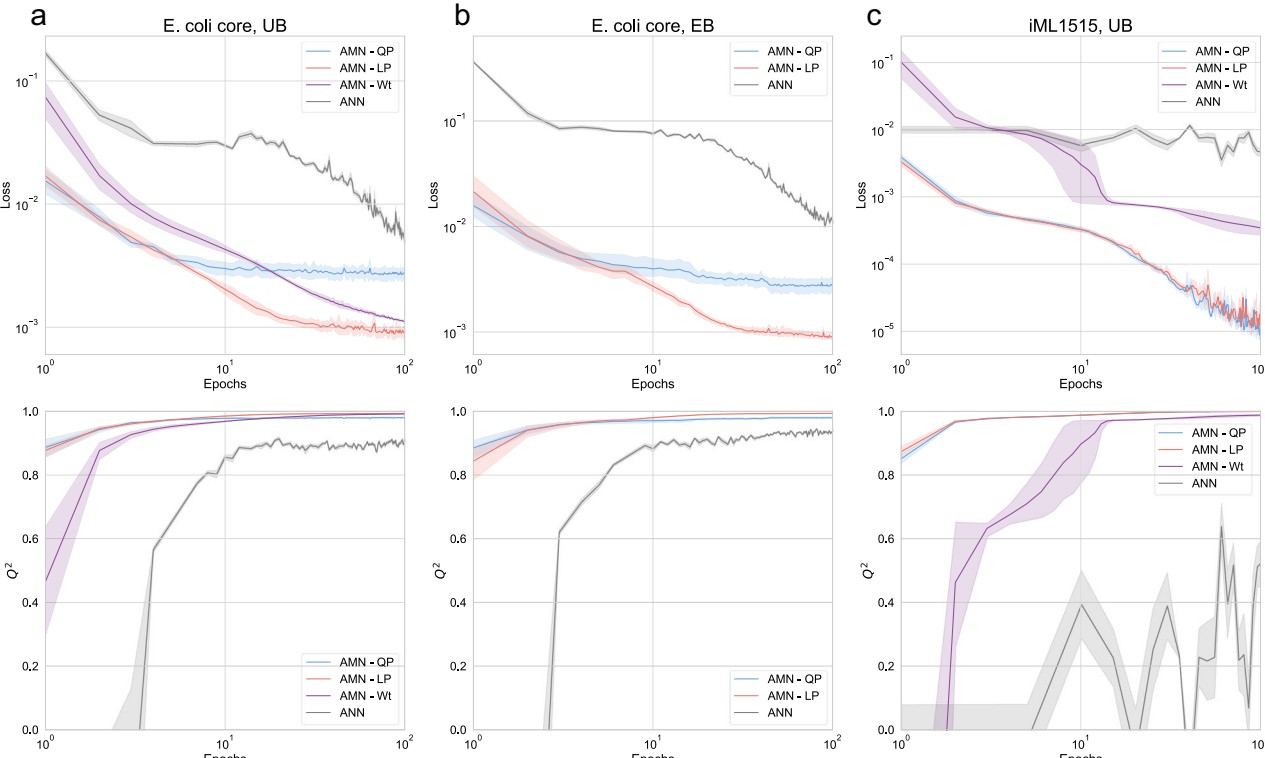

**Fig. 2 | Benchmarking AMNs with different training sets and mechanistic layers.** All results were computed on 5-fold cross-validation sets. Plotted is the mean and standard error (95% confidence interval) over the five validation sets of the cross-validation. Top panels show the custom mechanistic loss values, and bottom panels plot the $Q^2$ values for the growth rate, over learning epochs ($Q^2$ is the regression coefficient on cross-validation datapoints not seen during training). All AMNs have the architecture given in Fig. 1c, with $V_{in}$ as input, and a neural layer composed of one hidden layer of size 500. For all models, dropout = 0.25, batch size = 5, the optimizer is Adam, the learning rate is $10^{-3}$. The architecture for ANN (a classical dense network) is given in the "Methods" section it takes as input the uptake fluxes bounds $V_{in}$ and produce a vector $V_{out}$ composed of all fluxes with which the loss is computed. **a**–**c** show results for different training sets: **a**, **b** for 1000 simulations training sets generated with the *E. coli* core model, respectively with UB and EB as inputs, whereas **c** is for a 1000 simulations training set generated with the iML1515 model, with UB as input (for more details on the training set generations, refer to "Methods"). As mentioned in subsection "Alternative mechanistic models to surrogate FBA", AMN-Wt cannot be used to make predictions when exact bounds (EB) are used and is therefore not plotted in (**b**). Source data are provided as a Source Data file (cf. "Data availability").

architecture", a consensus set of weights leads to a solution when upper bounds (UB) for uptake fluxes are provided, but not when exact bounds (EB) for uptake fluxes are given (cf. Supplementary Fig. S2). Consequently, we cannot assume that the Wt-solver can handle all possible flux distributions in the EB case. To overcome this shortcoming, we next present two alternative methods that are much closer to the optimizations behind FBA and that can accommodate both EB and UB cases for uptake fluxes. The two methods address two distinct tasks in flux modeling: optimizing a flux distribution for maximal growth rate (LP-solver), as in classical FBA, and fitting a stationary flux distribution to partial flux data (QP-solver).

The second method (LP-solver), derived from a method proposed by Yang et al.[20], handles linear problems using exact constraint bounds (EBs) or upper bounds (UBs) for uptake fluxes ($V_{in}$). That method makes use of Hopfield-like networks, which is a long-standing field of research[21] inspired by the pioneering work of Hopfield and Tank[22]. As with the Wt-solver, the LP-solver iteratively computes fluxes to come closer to the steady-state solution ($V_{out}$). However, calculations are more sophisticated, and the method integrates the same objective function (e.g. maximize growth rate) as the classical FBA Simplex solver. The solver iteratively updates the flux vector, $V$, and the vector, $U$, representing the dual problem variables also named metabolites shadow prices[23] (cf. "LP-solver" in "Methods" and in Supplementary Information for further details).

The third approach (QP-solver), is loosely inspired by the work on physics-informed neural networks (PINNs), which has been developed

to solve partial differential equations matching a small set of observations[24]. With PINNs, solutions are first approximated with a neural network and then refined to fulfill the constraints imposed by the differential equations and the boundary conditions. Refining the solutions necessitates the computation of three loss functions. The first is related to the observed data, the second to the boundary conditions and the third to the differential equations. As detailed in "Methods", we similarly compute losses for simulated or measured reference fluxes, $V_{ref}$, the flux boundary constraints, and the metabolic network stoichiometry. As in PINN we next compute the gradient on these losses to refine the solution vector $V$. Unlike with the LP-solver, we do not provide an objective to maximize in the present case, but instead reference fluxes, consequently the method is named QP because it is equivalent to solving an FBA problem with a quadratic program.

To assess the validity of the LP and QP solver, we used the *E. coli* core model[25] taken from the BiGG database[26]. To generate with Cobrapy package[19] a training set of 100 growth rates varying 20 uptake fluxes, following the procedure given in "Methods". Results can be found in Supplementary Fig. S6, showing excellent performances after 10,000 iteration steps.

### AMNs: metabolic and neural hybrid models for predictive power with mechanistic insights

While the above solvers perform well, their main weakness is the number of iterations needed to reach satisfactory performances. Since

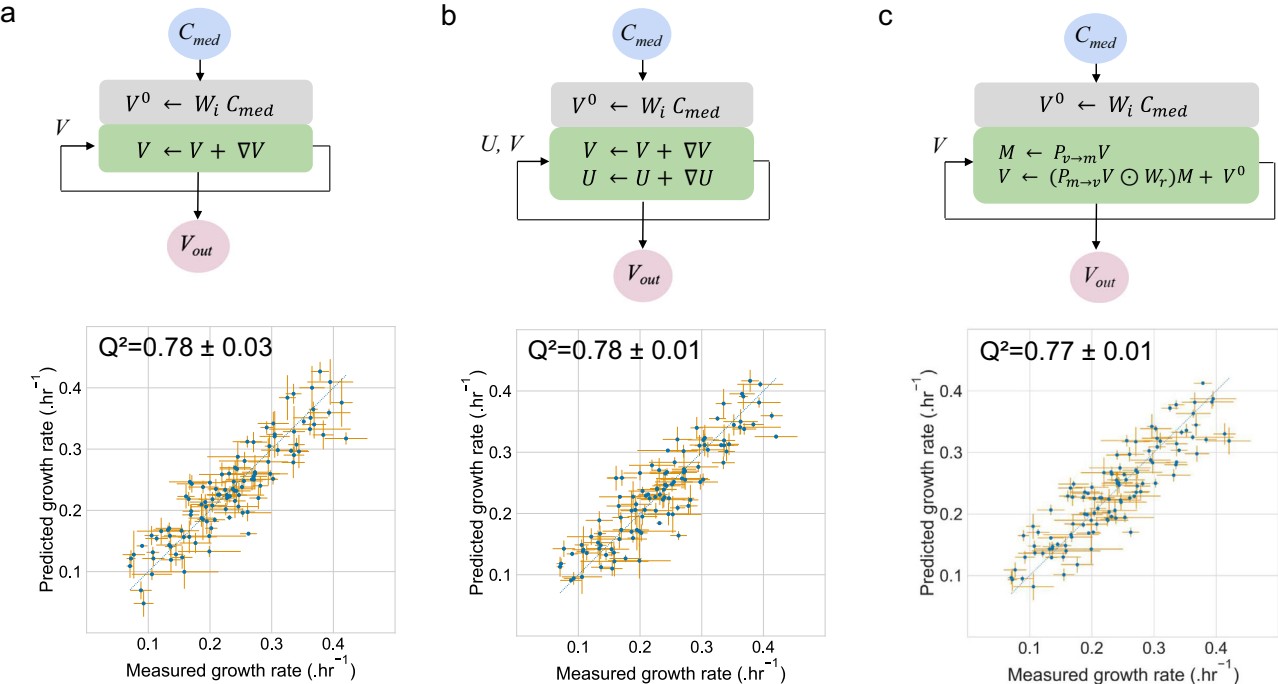

**Fig. 3 | Benchmarking growth rate predictions by AMNs with experimental measurements.** In all panels, the experimental measurements were carried out on *E. coli* grown in M9 with different combinations of carbon sources (strain DH5-alpha, model iML1515). Training and 10-fold stratified cross-validation were performed three times with different initial random seeds. All points plotted were compiled from predicted values obtained for each cross-validation set. In all cases, means are plotted for both axes (measured and predicted), and error bars are standard deviations. For the measured data, means and standard deviations were computed based on three replicates, whereas for predictions, means and standard deviations were computed based on the 3 repeats of the 10-fold cross-validation. **a** Architecture and performance of AMN-QP. The neural layer (gray box) is composed of an input layer of size 38 ($\mathbf{C_{med}}$), a hidden layer of size 500, and an output layer of size 550 corresponding to all fluxes ($\mathbf{V}$) of the iML1515 reduced model. The mechanistic layer (green box) follows the neural layer and minimizes the loss between measured and predicted growth rate, as well as the losses of the metabolic network constraints. The model was trained for 1000 epochs with dropout = 0.25, batch size = 5, and the Adam optimizer with a $10^{-3}$ learning rate. **b** Architecture and performance of AMN-LP. This model hyperparameters are identical to those of (**a**). The neural layer computes the initial values for the 550 reaction fluxes (vector V), the initial values for the 1083 metabolite shadow prices (vector U) are set to zero. **c** Architecture and performance of the AMN-Wt architecture. The model hyperparameters are those of the previous panels and the size of the $\mathbf{W_r}$ matrix is $550 \times 1083$ (sizes of $\mathbf{V}$ and $\mathbf{U}$ vectors). Source data are provided as a Source Data file (cf. "Data availability").

our goal is to integrate such methods in a learning architecture, this drawback has to be tackled. As illustrated in Fig. 1c, our solution is to improve our initial guesses for fluxes, by training a prior neural layer (a classical dense ANN) to compute initial values for all fluxes ($\mathbf{V_0}$) from bounds on uptake fluxes ($\mathbf{V_{in}}$) or media compositions ($\mathbf{C_{med}}$). This solution enables the training of all AMNs with few iterations in the mechanistic layer. In the remainder of the paper, we name AMN-Wt, AMN-LP and AMN-QP, the hybrid model shown in Fig. 1c composed of a neural layer followed by a mechanistic layer, i.e., a Wt, LP or QP solver.

The performances of all AMN architectures (Wt, LP, QP) and a classical ANN architecture (cf. Methods "ANN architecture", for further details) are given in Fig. 2, using FBA-simulated data on two different *E. coli* metabolic models, *E. coli* core[25] and iML1515[9]. These models are composed respectively of 154 reactions and 72 metabolites, and 3682 reactions and 1877 metabolites (after duplicating bi-directional reactions). In all cases, the training sets were generated by running the default Simplex-based solver (GLPK) of Cobrapy[19] to optimize 1000 growth rates for as many different media. Each medium was composed of metabolites found in minimal media (M9) and different sets of additional metabolites (sugars, acids) taken up by the cell (more details in Methods "Generation of training sets with FBA"). These training sets have as variables a vector of bounds on uptake fluxes (20 for *E. coli* core, 38 for iML1515) along with the Cobrapy[19] computed growth rate. For the ANN training set, to enable loss computation on constraints, we replaced the growth rate by the whole flux distribution computed by Cobrapy[19] (cf. Methods "ANN architecture").

Figure 2 shows the loss values on mechanistic constraints and the regression coefficient ($Q^2$) for the growth rates of the aforementioned models. All results shown are computed on 5-fold cross-validation sets. Additional information on hyperparameters and results on independent test sets are found in the Supplementary Information. In particular, Supplementary Fig. S7 shows performances obtained with AMN-QP and the *E. coli* core model with different neural layer architectures and hyperparameters, justifying our choices for the neural layers of AMNs (one hidden layer of dimension 500 and a training rate of $10^{-3}$). Similar results were found for AMN-LP and AMN-Wt. Additionally, in Supplementary Table S1, more extensive benchmarking is provided comparing MMs, ANNs and AMNs. This table shows performances for training, validation, and independent test sets of more diverse datasets, along with all training sets parameters and the models' hyperparameters.

All AMN architectures exhibit excellent regression coefficients and losses after a few learning epochs, and this for both models *E. coli* core[25] and iML1515[9]. It is interesting to observe the good performances of AMN-Wt when UB training sets are provided. Indeed, while counterexamples can be found for which AMN-Wt will not work with EB training sets (cf. Supplementary Fig. S2), we argue in the Supplementary Information "AMN-Wt architecture" that AMN-Wt is able to handle UB training sets because the initial inputs (UB values for uptake fluxes) are transformed into suitable exact bound values during training (via the neural layer).

We recall that in Fig. 2, ANNs were trained on all fluxes to enable loss computation (154 fluxes for *E. coli* core and 550 fluxes for iML1515),

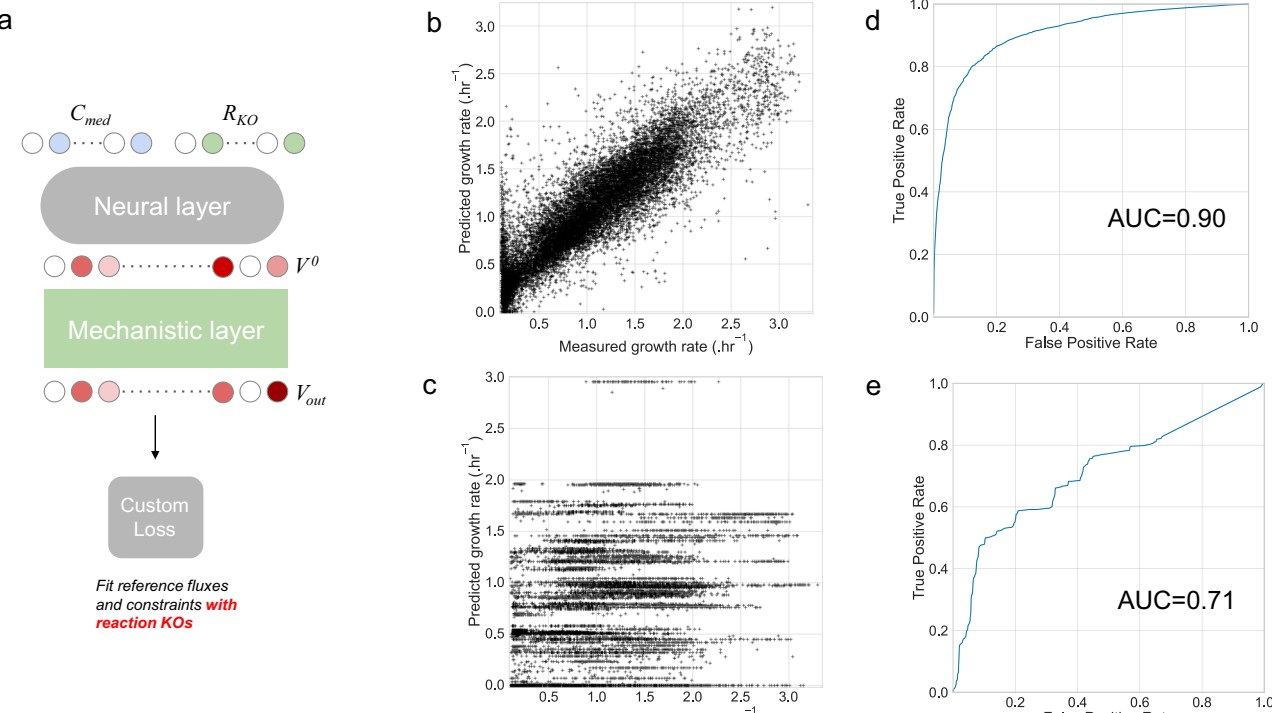

**Fig. 4 | AMNs growth rate predictions for *E. coli* gene KOs mutants.** An AMN model was trained on a set of 17,400 measured growth rates of *E. coli* grown in 120 unique media compositions and 145 different single metabolic gene KOs. **a** AMN architecture integrating metabolic gene KOs. This architecture is similar to Fig. 1c, except for a secondary input ($R_{KO}$) for the neural layer, alongside the medium composition $C_{med}$. The $R_{KO}$ input is a binary vector describing which reactions are KO. The custom loss function ensures that reference fluxes (i.e., the *E. coli* mutants measured growth rates) and mechanistic constraints are respected and that reactions experimentally KO have in $V_{out}$ a null flux value. The neural layer comprised one hidden layer of size 500 and the model was trained for 200 epochs with dropout = 0.25, batch size = 5, and the Adam optimizer with a $10^{-3}$ learning rate. **b** AMN regression performance on aggregated growth rate predictions from a 10-fold cross-validation. The mechanistic layer used for this architecture was the QP solver. **c** Regression performance of classical FBA with scaled upper bounds for compounds present in the medium and setting the upper bound and lower bound to zero for reactions that are KO (having a value of 0 in $R_{KO}$). **d** ROC curve of AMN results. We thresholded the measured growth rates (continuous values) in order to transform them into binary growth vs. no growth measures. **e** ROC curve of classical FBA results. The same thresholding as for (**d**) was applied. Source data are provided as a Source Data file (cf. "Data availability").

thus the number of training data points is substantially larger than for AMNs (154,000 or 550,000 for ANNs, instead of 1000 for AMNs). Despite requiring larger training sets, ANNs also need more learning epochs than AMNs to reach satisfying constraint losses and growth rate predictions for *E. coli* core (Fig. 2a, b) and do not handle well the large iML1515 GEM (i.e., the growth rate cannot be accurately predicted and the oscillatory behavior in Fig. 2c demonstrates 100 epochs are not enough to reach convergence).

## AMNs can be trained on experimental datasets with good predictive power

To train AMNs on an experimental dataset, we grew *E. coli* DH5-alpha in 110 different media compositions, with M9 supplemented with four amino acids as a basis and ten different carbon sources as possibly added nutrients. From 1 up to 4 carbon sources were simultaneously added to the medium at a concentration of 0.4 g l$^{-1}$ (more details in Methods "Culture conditions"). We determined which compositions to test by choosing all the 1-carbon source media compositions and randomly picking one hundred of the 2-, 3- and 4-carbon sources media compositions (more details in Methods "Generation of an experimental training set"). The growth of *E. coli* was monitored in 96-well plates, by measuring the optical density at 600 nm (OD$_{600}$) over 24 h. The raw OD$_{600}$ was then passed to a maximal growth rate determination method based on a linear regression performed on log(OD$_{600}$) data (more details in Methods "Growth rate determination").

The resulting experimental dataset of media compositions, $C_{med}$, and growth rates, $V_{ref}$, was used to train all AMN architectures (LP, QP, Wt). These architectures are those shown in Fig. 1c with $C_{med}$ as input. In all cases the mechanistic layer was derived from the stoichiometric matrix of the iML1515[20] *E. coli* reduced model (cf. Methods "Making metabolic networks suitable for neural computations"). Following Fig. 1c, $C_{med}$ was entered as a binary vector (presence/absence of specific metabolites in the medium), the vector was then transformed through the neural layer into an initial vector, $V^0$, for all reaction fluxes (therefore including the medium uptake fluxes) prior to be used in the mechanistic layer and the loss computations. Prediction performances are provided in Fig. 3, alongside schematics for each of the architectures.

For displaying meaningful results and to avoid any overfitting bias, we show in Fig. 3 predictions for points unseen during training. More precisely, we computed the mean and standard deviation of predictions over 3 repeats of stratified 10-fold cross-validations, each repeat having all points predicted, by aggregating validation sets predictions of each fold. Overall, results presented in Fig. 3 have been compiled over 3 × 10 = 30 different AMN models, each having different random seeds for the neural layer initialization and the train/validation splits.

As a matter of comparison, a decision tree algorithm predicting only the growth rate from $C_{med}$ (the RandomForestRegressor function from the sci-kit learn package[27] having 1000 estimators and other parameters left with default values) reach a regression performance of

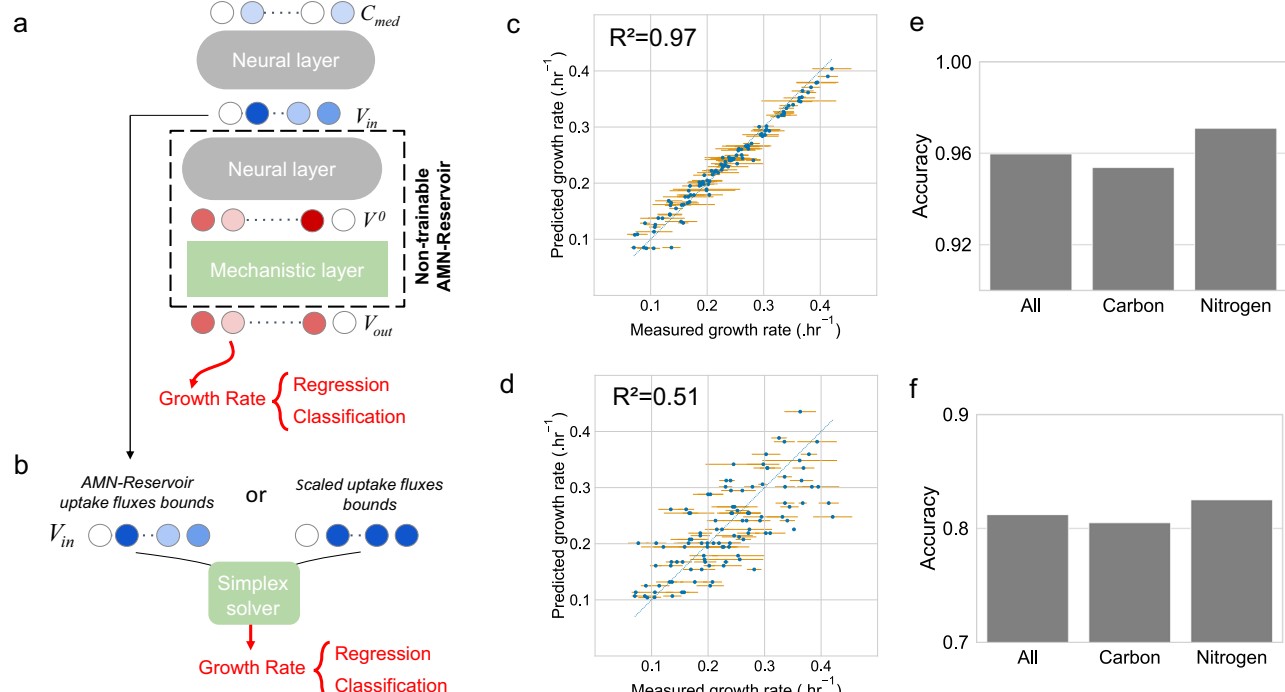

**Fig. 5 | Reservoir computing for improving the predictive power of FBA modeling (strain _E. coli_ DH5-alpha, model iML1515 and strain _P. putida_ KT2440, model iJN1463).** For (**c**, **d**), plotted are the measured growth rates means and standard deviations, computed from replicates (cf. "Methods"). **a** Learning architecture. The two-step learning is similar to what is shown in Fig. 1d. Here an AMN with QP-solver is trained either on iML1515 (**c**, **d**) or iJN1463 (**e**, **f**) with FBA simulations. The AMN (with frozen parameters) is then connected to a prior trainable network that computes medium uptake fluxes ($V_{in}$) from the medium composition ($C_{med}$). From $V_{in}$ the non-trainable reservoir returns all fluxes ($V_{out}$) including the growth rate. Next, a regression or classification is carried out on the growth rate. For results presented in (**c**, **e**), the neural layer comprised one hidden layer of size 500, the model was trained for 1000 epochs with dropout = 0.25, batch size = 5, and the Adam optimizer with a $10^{-4}$ learning rate. **b** Scheme showing the two possible inputs for Cobrapy (running a simplex solver), either using $V_{in}$ extracted from (**a**), or using scaled upper bounds on uptake fluxes. **c** Regression performance of Cobrapy for the _E. coli_ dataset when using $V_{in}$. **d** Regression performance of Cobrapy when using scaled upper bounds on corresponding uptake fluxes. **e** Accuracy performance of Cobrapy for the _P. putida_ dataset when using $V_{in}$. **f** Accuracy performance of Cobrapy when using original values of the study from Nogales et al. For the results in (**e**, **f**), accuracies are given for the whole dataset (All) composed of carbon source assays (Carbon) and nitrogen source assays (Nitrogen). Source data are provided as a Source Data file (cf. "Data availability").

$0.71 \pm 0.01$ with the same dataset and cross-validation scheme, indicating AMNs can outperform regular machine learning algorithms.

As one can observe in Fig. 3, the experimental variability on the measured growth rates is relatively high, and the $Q^2$ values could be interpreted differently if taking this variability into account. To study this further, we estimated the best possible $Q^2$ that can be reached at a given experimental variability. Precisely, for each experimental data point, we randomly drew a new point from a normal distribution with a mean and variance equal to what was experimentally determined for the original point. This point can be considered as an experimental randomized point. After doing this for all points and computing the $Q^2$, repeating this process 1000 times, we obtain a mean $Q^2 = 0.91$ with a standard deviation of 0.02. Consequently, the best possible $Q^2$ accounting for experimental variability is 0.91, and the performance of $Q^2 = 0.77$ (or 0.78) must be interpreted considering that value. Furthermore, substituting each point by a box defined by standard deviations of both measurement and prediction, we find that 79% for AMN-QP (76% for AMN-LP and 74% for AMN-Wt) of the boxes intersect the identity line indicating that these points are correctly predicted within the variances.

Our results show that AMNs can learn on FBA-simulated training sets and make accurate predictions while respecting the mechanistic loss, as shown in Fig. 2. AMNs can also perform well on a small experimental growth rates dataset as shown in Fig. 3. To demonstrate capabilities of AMNs beyond these tasks, we extracted from the ASAP database[28] a dataset of 17,400 growth rates for 145 _E. coli_ mutants. Each mutant had a KO of a single metabolic gene and was grown in 120 media with a different set of substrates. Our AMNs training set, were

therefore composed of medium composition and reaction KOs, both encoded as binary vectors, alongside the measured growth rates. More details can be found in Methods "External training sets acquisition". Results are presented in Fig. 4 and compared with classical FBA results, which were obtained running Cobrapy using the same dataset and setting scaled upper bounds (cf. Methods "Searching uptake fluxes upper bounds in FBA") corresponding to medium uptake fluxes and constraining KO reactions to zero fluxes in the metabolic model. The AMN architecture (Fig. 4a) used with this dataset is similar to the architecture shown in Fig. 1c, with an added input for reaction KOs ($R_{KO}$). Importantly, we also added a term to the custom loss in order to respect the reaction KOs (cf. Methods "Derivation of loss functions" for more details).

The AMN regression performance in Fig. 4 (aggregated predictions from a 10-fold cross-validation) reaches $Q^2 = 0.81$ (Fig. 4b). For comparison, a decision tree algorithm predicting only the growth rates from $C_{med}$ and $R_{KO}$ (the XGBRegressor function from the XGBoost package[29] with all parameters set to default values) yields a regression performance of 0.75, with the same cross-validation scheme and dataset.

The performance of classical FBA is poor, as no correlation can be found between measured and calculated growth rate (Fig. 4c). Such performance is expected as classical FBA relies on fixed uptake fluxes. In contrast, FBA should perform better to predict growth vs. no growth (a classification task), this is due to the fact that the network structure of GEMs already provides a lot of information on reaction essentiality, growth yields on different substrates, and other qualitative insights about metabolism. Indeed, in the most recent GEM of _E. coli_, iML1515[9],

an accuracy >90% was found for a dataset of growth assays based on media compositions (classification task predicting growth vs. no growth). Consequently, we also show in Fig. 4 the performances of AMN and FBA for classification. Precisely, we treat the growth rate value predicted by either the AMN or FBA as a classification score for growth vs. no growth, to that end and following Orth et al.[30], we threshold the growth rate measures (continuous values) to binary values (1 when the growth rate is above 5% of the maximum growth rate of the dataset, 0 otherwise). With classifications, one can compute ROC curves, and these are shown in Fig. 4d for AMN and Fig. 4e for classical FBA.

Overall, the results presented in Fig. 4 show that for both regression and classification tasks, AMNs, which integrates learning procedures, outperforms classical FBA, which is based on maximizing a biological objective only. Indeed, as mentioned in the introduction, one main issue of classical FBA is the unknown uptake fluxes, which have a large impact on the predicted growth rate value, while AMNs can handle this problem because of their learning abilities. To further showcase AMN capabilities, in particular when multiple fluxes are measured, we provide in Supplementary Fig. S8 the performance of an AMN on a dataset from Rijsewijk et al.[31]. With this dataset, composed of 31 fluxes measured for 64 single regulator gene KO mutants of *E. coli* grown in 2 media compositions, our AMN reaches a variance averaged $Q^2$ value of 0.91 in 10-fold cross-validation.

### AMNs can be used in a reservoir computing framework to enhance the predictive power of traditional FBA solvers

As already mentioned in the introduction section, the uptake fluxes of *E. coli* nutrients, as well as their relation to external nutrient concentrations, remain largely unknown: the uptake flux for each compound may vary between growth media. In classical FBA calculations, this is usually ignored and the same upper bound (or zero, if a compound is absent) is used in all cases. Our results for KO mutants suggest that this strongly reduced regression performance of classical FBA, while in classification the effect is less severe. Nonetheless, for regression or classification the problem remains: how can realistic uptake fluxes be found?

In the following, we show a way to find these uptake flux values and improve the performances of classical FBA solvers (for both regression and classification). Once an AMN has been trained on a large dataset of FBA-simulated data, we can fix its parameters and exploit it in subsequent further learning in order to find uptake flux values that can be used in a classical FBA framework. Loosely inspired by reservoir computing[32], we call this architecture "AMN-Reservoir" (Figs. 1d and 5a). Let us note that we are not using usual reservoirs[32] with random weights and a post-processing trainable layer. As a matter of fact, we do not reach satisfactory performances when we substitute the AMN-Reservoir weights (learned during training) by random weights.

We benchmarked our AMN-Reservoir approach with two datasets. The first one is the one used in Fig. 3, composed of 110 *E. coli* growth rates, and the second is a growth assay of *P. putida* grown in 296 different conditions[33] (more details in Methods "External training sets acquisition"). The procedure used for the two datasets is the same. First, the AMN-Reservoir is trained on FBA simulations. For *E. coli* we used as an AMN-Reservoir the AMN-QP of Supplementary Table S1 trained on an iML1515 UB dataset, for *P. putida* we used the AMN-QP of Supplementary Table S1 trained on an iJN1463[33] UB dataset. Second, as shown in Fig. 5a, the whole experimental dataset is used to train the neural layer, setting up either a regression task for *E. coli* growth rates and a classification task for *P. putida* growth assays (growth vs. no-growth). After training on media compositions and measured growth rates (for both *E. coli* and *P. putida*), we extract the corresponding uptake fluxes ($V_{in}$). These uptake fluxes are then taken as input for a classical FBA solver for growth rate calculation, as shown in Fig. 5b.

The output of FBA was used to produce the results shown in panels c and e. As a matter of comparison, we show the performance of FBA for the *E. coli* dataset (Fig. 5d) with scaled uptake fluxes bounds (cf. Methods "Searching uptake fluxes upper bounds in FBA"), and for *P. putida* (Fig. 5f) where we used the same flux bounds as given in the reference study[33].

Overall, results shown in Fig. 5 indicate that the usage of AMN-Reservoirs substantially increases the predictive capabilities of FBA without additional experimental work. Indeed, after applying the AMN-Reservoir procedure to find the best uptake fluxes, we raised the $R^2$ on *E. coli* growth rates from 0.51 (panel d) to 0.97 (panel c) and we raised the accuracy on *P. putida* growth assays from 0.81 (panel f) to 0.96 (panel e). We note that these uptake fluxes were found for the training set of the AMN-Reservoir, but we also show the performance of FBA with uptake fluxes found for cross-validation sets (Supplementary Fig. S9). As expected, Fig. S9 displays the same level of performance as the AMNs directly trained on experimental data (Fig. 3).

## Discussion

In this study we showed how a neural network approach, with metabolic networks embedded in the learning architecture, can be used to address metabolic modeling problems. Previous work on RNNs and PINNs for solving constrained optimization problems was re-used and adapted to develop three models (AMN-Wt, -LP and -QP) enabling gradient backpropagation within metabolic networks. The models exhibited excellent performance on FBA generated training sets (Fig. 2 and Supplementary Table S1). We also demonstrated that the models can directly be trained on an experimental *E. coli* growth rate dataset with good predictive abilities (Fig. 3).

In classical FBA, all biological regulation mechanisms behind a flux distribution are ignored and flux computation relies entirely on bounds set on uptake or internal fluxes. Therefore, when performing classical FBA, one needs to set uptake bounds individually for each condition to reliably predict metabolic phenotypes. AMNs attempt to capture the overall effects of regulation via the neural layer while keeping the mechanistic layer for the metabolic phenotype. Indeed, as shown in Fig. 4 and Supplementary Fig. S8, gene KOs of metabolic enzymes or regulators can be taken into account via the neural layer. Such AMNs can potentially be trained on a variety of experimental inputs (wider than the carbon source composition shown in our studies) to grasp the effects of complex regulation processes in the cell and to better explain the end-point metabolic steady-state phenotype of an organism.

For improved adaptability, we also trained AMN-Reservoirs on large FBA-simulated training sets and used these to improve FBA computations on two experimental datasets (*E. coli* growth rates and *P. putida* growth assays). Figure 5 shows that our hybrid models substantially enhance classical FBA predictions both quantitatively and qualitatively, and this without any additional flux measurements.

One issue that impairs phenotype predictions with FBA is the lack of knowledge on media uptake fluxes and determining bounds on these fluxes is a core experimental work required for making classical FBA computations realistic. These bounds depend on cell transporters abundances, which may vary between conditions and depend on the cell's metabolic strategy. satFBA[34] is a variant of FBA that assumes fixed transporter levels and converts medium concentrations to possible uptake fluxes by kinetic rate laws, relying on a Michaelis-Menten value for each uptake reaction. In more sophisticated CBM approaches, such as molecular crowding FBA[35] or Resource Balance Analysis[36], constraints on the resource availability and allocation are added to obtain more biologically plausible metabolic phenotypes, but parameterizing such models requires additional data. To provide the necessary data to the aforementioned CBM methods and to validate results, fluxomics[37], metabolomics[38], or transcriptomics[39] have been used in the past. Because additional experimental work is needed with sophisticated

CBM approaches, many users rely on classical FBA, which as we have seen, has limitations as far as quantitative predictions are concerned.

AMNs are used in this study for tackling the same issue as satFBA: predicting metabolites uptake fluxes from medium metabolite composition. To do so, where satFBA uses transporter kinetics with parameters that need to be acquired through additional measurements, AMNs use a pre-processing neural layer that is accessible for learning. Our AMN hybrid models get rid of additional experimental data for reaching plausible fluxes distributions. We do so by backpropagating the error on the growth rate or any other measured flux, to find complex relationships between the medium compositions and the medium uptake fluxes. To this end, we demonstrated the high predictive power of AMNs, and their re-usability in classical FBA approaches. Indeed, FBA developers and users may now make use of our AMN-Reservoir method for relating medium uptake fluxes to growth medium compositions. In this regard, a Source Data file (cf. Data availability) gives uptake fluxes for the metabolites used in our benchmarking work with *E. coli* and *P. putida* (Fig. 5), and these upper bounds for uptake fluxes can directly be used by Cobrapy to reproduce Fig. 5c, e.

Making FBA suitable for machine learning as we have done in this study opens the door to improve GEMs. For instance, in addition to estimating uptake fluxes, AMNs could be used to estimate the coefficients of the biomass reaction based on measurements. So far, these coefficients are derived based on literature, but also using experimental data: growth rate, fluxes, and macromolecular fractions measures can help finding optimal coefficients[9]. However, these experiments are limited in number, and biomass coefficients are usually determined only once, for a single experimental setup, and are hardly extrapolated to all possible conditions. Some studies already underline this issue and attempt to efficiently integrate experimental data in the biomass reaction parametrization[40]. With AMNs, a trainable layer containing the biomass coefficients could be added, adapting the biomass reaction to any experimental setup. Another possible application of AMN is to enhance GEMs reconstruction based on quantitative prediction performance. Indeed, the method we developed for KOs could be adapted to screen putative reactions in a metabolic model so that its predictions match experimental data. This task should be performed after a manual curation, of course, to rely on existing literature knowledge and databases.

Returning to the curse of dimensionality issue mentioned in the introduction, we systematically studied at which training set sizes 'black-box' ML methods would yield performances similar to our AMN hybrid models. To that end, we trained a simple dense ANN model on training sets of increasing sizes. Results obtained with *E. coli* core[25] show that at least 500,000 labeled data (reference fluxes) are needed in the training sets to reach losses below 0.01 (cf. in Supplementary Fig. S10), which according to Fig. 2 and Supplementary Table S1 are still one order of magnitude higher than all AMNs losses trained on only 1000 labeled data. This clearly demonstrates the capacity of hybrid models to reduce training set sizes by constraining the search space through the mechanistic layer. Other black-box models can also be used, indeed the experimental measurements used in Figs. 3 and 4 can be fitted with decision tree algorithms (Random Forests[27] and XGBoost[29]) with performances slightly under those of AMN. However, with these algorithms, nothing is learned regarding the mechanistic constraints and results produced by these methods cannot be fed back to classical FBA, as we do in Fig. 5 with the AMN-Reservoir.

Beyond improving constraint-based mechanistic models and black-box ML models, AMNs can also be exploited for industrial applications. Indeed, since arbitrary objective functions can be designed and AMNs can be directly trained on experimental measurements, AMNs can be used to optimize media for the bioproduction of compounds of interest or to find optimal gene deletion and insertion strategies in typical metabolic engineering projects. In this latter case, reactions would be turned off via a trainable layer, which would be added prior to the mechanistic layers of our AMNs. Another potential application is the engineering of microorganism-based decision-making devices for the multiplexed detection of metabolic biomarkers or environmental pollutants. Here, AMNs could be used to search for internal metabolite production fluxes enabling one to differentiate positive samples containing biomarkers or pollutants from negative ones. Such a device has already been engineered in cell-free systems[41], and AMNs could be used to build a similar device in vivo by adding a trainable layer after the mechanistic layer whose purpose would be to select metabolite production fluxes that best split positive from negative samples.

## Methods

### Making metabolic networks suitable for neural computations
The set-up of our AMNs requires all reactions to be unidirectional; that is, the solutions must show positive-only fluxes (which is not guaranteed by usual GEMs). To split reactions of a given metabolic network into separate forward and reverse reactions, we wrote a standardization script that loads an SBML model into Cobrapy[19] and screens for all two-sided reactions, then duplicating them into two separate reactions; and writes a new version of the model with bi-directional reactions split into separate forward and backward reactions. To avoid confusion, we add a suffix to these reaction names, either "for" or "rev" respectively designating the original forward reaction and the reversed reaction. The uptake reactions were also duplicated, even if encoded as one-sided, and their suffix was set to "i" for inflow reactions (adding matter to the cell), and "o" for outflow reactions (removing matter from the system).

As detailed in the next subsection, our unidirectional models are used to build flux data training sets. The duplicated iML1515[9] model is large, comprising 3682 reactions and 1877 metabolites. A substantial number of reactions in this model have zero fluxes for many different media, and it is unnecessary to keep these reactions during the training process. Prior to training, we therefore generated a reduced model by removing reactions having zero flux values along with the metabolites no longer contributing to any reactions. Using that procedure, we were able to reduce iML1515[9] model to only 550 reactions and 1083 metabolites.

### Generation of training sets with FBA
Our reference flux data were obtained from FBA simulations, using the GNU Linear Programming Kit (GLPK, a simplex-based method) on Cobrapy[19], with different models of different sizes. Throughout this paper, when "reference FBA-simulated data" is mentioned, it refers to data computed with this method.

Reference FBA-simulated data for metabolic flux distributions were generated using models downloaded from the BiGG database[26]. The models were used to generate data using Cobrapy[19] following a precise set of rules. First, we identified essential uptake reactions for the models we used (*E. coli* core[25] and iML1515[9]) which we defined in the following way: if one of these reactions has its flux upper bound set to 0 mmol gDW$^{-1}$ h$^{-1}$, the biomass reaction optimization is impossible, even if all other uptake fluxes bounds are set to a high value, e.g., 1000 mmol gDW$^{-1}$ h$^{-1}$. In other words, we identified the minimal uptake fluxes enabling growth according to the models. For *E. coli* core[25] we found seven of such obligate reactions (for the uptake of $CO_2$, $H^+$, $H_2O$, $NH_4$, $O_2$, Phosphate, and Glycerol as the carbon source). For iML1515[20] we had the same 7 obligate reactions and additional salt and ions uptake reactions (for the uptake of $Fe^{2+}$, $Fe^{3+}$, $Mn^{2+}$, Zinc, Mg, Calcium, $Ni^{2+}$, $Cu^{2+}$, Selenate, $Co^{2+}$, Molybdate, Sulfate, $K^+$, Sodium, Chloride, Tungstate, Selenite). With iML1515[9], we also added as obligate reactions the uptake of four amino acids (Alanine, Proline, Threonine and Glycine) in order to be consistent with our experimental training set where the four amino acids were systematically

added to M9 (cf. subsection "Generation of an experimental training set"). During reference FBA-simulated data generation, the upper bounds on these obligate reactions were set to 10 mmol gDW$^{-1}$ h$^{-1}$.

To generate different media compositions, we added to the obligate reactions a set of variable uptake reactions. For the *E. coli* core model[25] we added 13 variable uptake reactions (for Acetate, Acetaldehyde, Oxoglutarate, Ethanol, Formate, Fructose, Fumarate, Glutamine, Glutamate, Lactate, Malate, Pyruvate, and Succinate). For each generated medium, a set of variable uptake reactions was selected, drawn with a binomial distribution B($n$, $p$) with $n = 13$ and $p = 0.5$, $p$ being a tunable parameter related to the ratio of selected reaction. Consequently, the mean number of selected variable uptake reactions was $n \times p = 6.5$. Next for each selected reaction, the upper bound continuous value of the reaction flux was randomly drawn from a uniform distribution between 2 and 10 mmol gDW$^{-1}$ h$^{-1}$. For the iML1515[9] model, to limit the combinatorial search space, the selected variable uptake reactions were those of the experimental training set and consequently between 1 and 4 variable uptake reaction were added (cf. subsection "Generation of an experimental training set"). The upper bound values for each selected variable reaction were chosen randomly between 0 and 2.2 mmol gDW$^{-1}$ h$^{-1}$ (0 excluded). The 2.2 threshold was chosen to produce predicted growth rates that were in the range of those observed experimentally. For the *P. putida* iJN1463[33] model, we used the same approach with variable uptake reactions selected from the experimental training set, and consequently 1 variable uptake reaction was added to obligate reactions (described as the minimal medium in the reference study[33]) for each element of the training set. The upper bound values for each selected variable reaction were chosen randomly between 0 and 10 mmol gDW$^{-1}$ h$^{-1}$ (0 excluded).

After generating the set of growth media for *E. coli* core[25], iML1515[9] and iJN1463[33] we ran FBA in Cobrapy[19] for each medium and recorded all steady-state fluxes including the growth rate (flux of the biomass reaction). These fluxes were used as a training set for all models presented in Fig. 2 and in Supplementary Table S1. All AMN architectures were trained on the biomass flux (i.e. the growth rate), while ANN architectures were trained on all fluxes. For all UB training sets, the variable uptake flux values were those used by Cobrapy[19] to generate the training set. For EB training sets, the variable uptake flux values were those calculated by Cobrapy[19] at steady state.

### Derivation of loss functions

Loss functions are necessary to assess the performances of all MM solvers and all AMN architectures (AMN-Wt, -QP, and -LP) and also to compute the gradients of the QP solvers. In the following and subsequent subsections, all vectors and matrices notations are defined when they are first used and can also be found in Supplementary Table S2.

To compute loss, we consider a metabolic model with $n$ reactions and $m$ metabolites. Let $\boldsymbol{V} = (v_1, \ldots, v_n)^T$ be the reaction flux vector and $\boldsymbol{S}$ the $m \times n$ stoichiometric matrix of the model. We assume some metabolites can be imported in the model through a corresponding uptake reaction. Let $\boldsymbol{V_{in}}$ be the vector of $n_{in}$ upper bounds (or exact values) for these uptake reactions, and let $\boldsymbol{P_{in}}$ the $n_{in} \times n$ projection matrix such that $\boldsymbol{V_{in}} = \boldsymbol{P_{in}V}$. We further assume that some reaction fluxes have been experimentally measured, let $\boldsymbol{V_{ref}}$ be the vector of reference flux data (FBA-simulated or measured). With $\boldsymbol{P_{ref}}$ the $n_{ref} \times n$ projection matrix for measured fluxes. $\boldsymbol{V}$ is calculated by solving the following quadratic program (QP):

$$\min(|\boldsymbol{P_{ref}V} - \boldsymbol{V_{ref}}|^2)$$
$$\text{s.t. } \boldsymbol{SV} = 0$$
$$\boldsymbol{P_{in}V} \leq \boldsymbol{V_{in}} \quad (1)$$
$$\boldsymbol{V} \geq 0$$

For each solution, $\boldsymbol{V}$, of Eq. (1), four loss terms are defined. $L_1$ is the loss on the fit to the reference data. $L_2$ ensures the respect of the network stoichiometric constraint ($\boldsymbol{SV} = 0$). $L_3$ ensures the respect of the constraints on input fluxes that depend on medium composition ($\boldsymbol{P_{in}V} \leq \boldsymbol{V_{in}}$). Finally, $L_4$ ensures the respect of the flux positivity ($\boldsymbol{V} \geq 0$). The losses are normalized, respectively by $n_{ref}$ for the fit to reference data, $m$ for the stoichiometric constraint, $n_{in}$ for the boundary constraints, and $n$ for the flux positivity constraints.

Summing the four terms, the loss $L$ is:

$$\begin{aligned} L &= L_1 + L_2 + L_3 + L_4 \\ &= \tfrac{1}{n_{ref}}|\boldsymbol{P_{ref}V} - \boldsymbol{V_{ref}}|^2 + \tfrac{1}{m}|\boldsymbol{SV}|^2 + \tfrac{1}{n_{in}}|ReLU(\boldsymbol{P_{in}V} - \boldsymbol{V_{in}})|^2 + \tfrac{1}{n}|ReLU(-\boldsymbol{V})|^2 \end{aligned} \quad (2)$$

More details about each loss term can be found in the Supplementary Information "QP-solver equations".

When reaction KOs are added to the input of AMNs (as in Fig. 4), we add a term to the loss function, $L_5$, for ensuring a null value for fluxes that have their reaction KO:

$$L_5 = \frac{1}{n_{KO}}|ReLU(\boldsymbol{P_{KO}V} - \boldsymbol{R_{KO}})|^2 \quad (3)$$

where $\boldsymbol{R_{KO}}$ is a vector of length $n_{KO}$ describing which reactions are KO, and $\boldsymbol{P_{KO}}$ the projection matrix mapping the whole flux vector $\boldsymbol{V}$ to KO fluxes.

### Wt-solver

The Wt-solver describes a metabolic state by two vectors $\boldsymbol{V}$ and $\boldsymbol{M}$, representing respectively the reaction fluxes and the metabolite production fluxes. The initial value ($\boldsymbol{V^0}$) for vector $\boldsymbol{V}$ can be arbitrary as long as the uptake medium bounds are respected. Vectors $\boldsymbol{V}$ and $\boldsymbol{M}$ are iteratively computed until convergence using the following equations:

$$\begin{aligned} \boldsymbol{M} &= \boldsymbol{P_{v \to m}V} \\ \boldsymbol{V} &= (\boldsymbol{P_{m \to v}V} \propto \odot \boldsymbol{W_r})\boldsymbol{M} + \boldsymbol{V^0} \end{aligned} \quad (4)$$

where $\boldsymbol{W_r}$ is a consensus weight matrix representing flux branching ratios, $\boldsymbol{P_{v \to m}} = ReLU(\boldsymbol{S})$, $\boldsymbol{P_{v \to m}} = ReLU\left(\left[\frac{-1}{z_i s_{j,i}}\right]\right)$, $\boldsymbol{S}$ is the stoichiometric matrix, $s_{j,i}$ the stoichiometric coefficient of row $j$ and column $i$, $z_i$ the number of strictly negative elements in column $i$ of $\boldsymbol{S}$, and $\odot$ the Hadamard product operation. Additional details on the procedure and the associated matrices are provided in Supplementary Information section "Wt-solver equations".

### LP-solver

The LP method aims at solving linear constrained problem similar to the ones solved by FBA. It relies on the results from Yang et al.[20] where the authors used gradient descent on both primal and dual variables of the problem.

When the uptake fluxes are known (EB method), the FBA problem can be written as:

$$\begin{aligned} \max &: c_{FBA}^T \boldsymbol{V} \\ \text{s.t. } &\boldsymbol{S_{int}V} = -b_{FBA} \\ &\boldsymbol{V} \geq 0 \end{aligned} \quad (5)$$

where $\boldsymbol{S_{int}}$ is the stoichiometric matrix with uptake fluxes zeroed out (i.e. fluxes that add matter in the system). In other words, $\boldsymbol{S_{int}}$ is the internal stoichiometric matrix. Let us consider $\boldsymbol{b_{FBA}}$, a vector of dimension $m$ with $b_i$ corresponding to uptake fluxes of medium metabolite $m_i$ (either as an exact value for EB or an upper bound for UB) and $\boldsymbol{c_{FBA}}$, the objective vector of dimension $n$ (in this work this vector has non-zero elements only for reference fluxes like the biomass reaction flux, i.e., the growth rate.

This problem can be written in its dual form with $U$ being the dual variable of $V$:

$$\min : -b^T_{FBA} \, U$$
$$\text{st} : S^T_{int} U \leq c_{FBA} \tag{6}$$

As mentioned before, the problem given by Eq. (5) can be solved conjointly with problem given by Eq. (6) by iteratively updating $V$ and its dual $U$ through gradient descent:

$$V^{(t+1)} = V^{(t)} - dt \, \nabla V$$
$$U^{(t+1)} = U^{(t)} - dt \, \nabla U \tag{7}$$
$$V^{(0)} = P^T_{in} V_{in} \text{ and } U^{(0)} = 0$$

where $t$ is the iteration number and $dt$ the learning rate.

Note that initialization of LP with uptake fluxes is not mandatory with the method from Yang et al.[20] as it has been proven to converge to global optimum independently from the initial values of $V$ and $U$. Detailed expressions and derivations of gradients for $U$ and $V$ are provided in Supplementary Information "LP-solver equations" along with Figs. S4 and S5.

## QP-solver
The QP solver solves the quadratic program given by Eq. (1). While the QP system can be solved by a simplex algorithm, solutions can also be approximated by calculating the vector $V$ that minimizes the loss ($L$ in Eq. (2)). The gradient $\nabla V$ for vector $V$ can thus be found by solving $\frac{\partial L}{\partial V} = 0$ and, as in Eq. (7), $V$ is computed iteratively with iteration number $t$ and learning rate $dt$.

Detailed expressions and derivations for the gradient $\nabla V$, when exact bounds (EBs) or upper-bounds (UBs) are provided for uptake flux medium, can be found in the Supplementary Information "QP-solver equations".

## ANN architecture
The ANN architecture is a "black box" dense neural network. As with the other architectures the input layer corresponds to the medium uptake fluxes, $V_{in}$, and the output layer corresponds to the set of all fluxes $V_{out}$. In order to assess losses with the ANN architecture, which does not have any mechanistic layer, each entry of the training set contained all flux values (in other words, $V_{ref}$ contains all fluxes). Consequently, the training process with ANN consists in fitting all predicted fluxes to reference flux data (computing the MSE on all the fluxes). To compare results with the other architectures, $R^2$ and $Q^2$ are computed for the growth rate, and constraint losses are computed using predictions for all fluxes, using the formulation given in the subsection Methods "Derivation of loss functions".

## AMN architectures
As shown Figs. 1 and 3, we propose three AMN architectures: AMN-Wt, AMN-LP and AMN-QP. The AMNs are run with training sets using exact values (EB) or only upper bound values (UB) for medium uptake fluxes. All AMNs take as their input a vector of bounds of size $n_{in}$ for medium uptake fluxes ($V_{in}$) and then transform it via a dense neural network the input vector into an initial vector of size $n$ for all fluxes ($V^0$), which is refined through an iterative procedure computing $V^{(t+1)}$ from $V^{(t)}$. With all AMNs a $n_{in} \times n$ weight matrix transforming $V_{in}$ to $V^0$ is learned during training, and we name this transforming layer the neural layer. With AMN-LP/QP, $V^{(t)}$ is iteratively updated in a mechanistic layer by the gradient ($\nabla V$) of LP/QP solvers (cf. previous subsections in "Method"). With AMN-Wt, the mechanistic layer computes $V^{(t+1)}$ from $V^{(t)}$ using the transformations shown in Fig. S1, which include a $n \times m$ weight matrix ($W_r$). That weight matrix can be directly computed from training data when all fluxes are provided or can be learned during

training, when only a fraction of fluxes are provided (like the growth rate with experimental datasets). In our implementation (cf. "Code availability" section) AMN-Wt is embedded in a RNN Keras cell[42] and both matrices $W_i$ and $W_r$ are learned during training. ANN-Wt is further detailed in Supplementary Information "AMN-Wt architecture".

With all AMN architectures, the values of $V$ corresponding to $V_{in}$ are not updated in the neural nor mechanistic layers when training with exact values for medium uptake (EB training sets).

## ANN and AMN training parameters
For ANN and AMN architectures, we use the mean squared error ($L_1$ in Eq. (2)) for measured fluxes as the objective function to minimize during training. In all AMN architectures we add to the $L_1$ loss function the terms corresponding to the 3 losses derived from the constraints of the metabolic model ($L_2$, $L_3$ and $L_4$ in Eq. (2)).

The parameters used when training ANNs and AMNs, there are two types:
(1) Reference data parameters: reference data can either be FBA-simulated or experimental. For FBA-simulated data, we can tune the size of the training set to be generated. We can also modify the mean number of selected variable intake medium fluxes, and the number of levels (i.e. the resolution) of the fluxes. We can also modify the variable uptake reactions list, but this modifies the architecture of the model (initial layer size), so we kept the same list for each model in the present work. The lists can be found in the subsection "Generation of training sets with FBA".
(2) Model hyperparameters: during learning on FBA-simulated or experimental data, ANN and AMN have a small set of parameters to tune: the number and size of hidden layers, the number of epochs, the batch size, the dropout ratio, the optimizer and learning rate, and the number of folds in cross-validation. These numbers are provided in Supplementary Table S1 for models trained on FBA-simulated data and in the captions of Figs. 3–5 for models trained on experimental data.

## Searching uptake fluxes upper bounds in FBA
The goal of this optimization was to find the best scaler for fluxes to best match experimentally determined growth rates, by using "out-of-the-box" FBA, simply informing the presence or absence of the flux according to the experimental medium composition. The optimal scaler used in Figs. 4 and 5 was found using the Cobrapy software package[19] by simply searching for the maximum $R^2$ between experimental and FBA-predicted growth rates for scalers ranging between 1 and 10.

## Generation of an experimental training set
Ten carbon sources were picked for being the variables of our training sets: Ribose (Sigma-Aldrich, CAS:50-69-1), Maltose (Sigma-Aldrich, CAS:6363-53-7), Melibiose (Sigma-Aldrich, CAS:585-99-9), Trehalose (Sigma-Aldrich, CAS:6138-23-4), Fructose (Sigma-Aldrich, CAS:57-48-7), Galactose (Sigma-Aldrich, CAS:59-23-4), Acetate (Sigma-Aldrich, CAS:127-09-3), Lactate (Sigma-Aldrich, CAS:867-56-1), Succinate (Sigma-Aldrich, CAS:150-90-3), Pyruvate (Sigma-Aldrich, CAS:113-24-6). These could ensure observable growth as a sole carbon source with a concentration of 0.4 g $l^{-1}$ in our M9 preparations. The selected carbon sources enter different parts of the metabolic network: 6 sugars enter the upper glycolysis pathway, and 4 acids enter the lower glycolysis pathway or the TCA cycle. With a binary (i.e., presence or absence of each carbon source) approach when generating the combinations to test for making the experimental training set, we generated all possible combinations of 1, 2, 3 or 4 carbon sources simultaneously present in the medium. Naturally, we picked all 1-carbon source media combinations for experimental determination (only 10 points). Then, we randomly selected 100 more combinations to experimentally determine, by randomly picking 20 points from the 2-, 40 points from the 3- and

40 points from the 4-carbon source combinations sets. The python scripts to generate these combinations and pick the ones for making our experimental training set are available on our Github package[43] (cf. "Codes availability" section). After picking the combinations to test, we experimentally determined the maximum specific growth rate of *E. coli* for each combination of carbon sources in M9 (cf. next two subsections). The mean over replicates for each media composition was computed as the corresponding growth rate value to make the final experimental training set (cf. Methods "Growth rate determination").

## Culture conditions

The base medium for culturing *E. coli DH5-α* (DH5a) was a M9 medium prepared with those final concentrations: 100 μM CaCl₂ (Sigma-Aldrich, CAS:10035-04-8); 2 mM MgSO₄ (Sigma-Aldrich, CAS:7487-88-9); 1X M9 salts: 3 g l⁻¹ KH₂PO₄ (Sigma-Aldrich, CAS: 7778-77-0), 8.5 g l⁻¹ Na₂HPO₄ 2H₂O (Sigma-Aldrich, CAS:10028-24-7), 0.5 g l⁻¹ NaCl (Sigma-Aldrich, CAS:7647-14-5), 1 g l⁻¹ NH₄Cl (Sigma-Aldrich, CAS:12125-02-9); 1X trace elements: 15 mg l⁻¹ Na₂EDTA 2H₂O (Sigma-Aldrich, CAS:6381-92-6), 4.5 mg l⁻¹ ZnSO₄ 7H₂O (Sigma-Aldrich, CAS:7446-20-0), 0.3 mg l⁻¹ CoCl₂ 6H₂O (Sigma-Aldrich, CAS:7791-13-1), 1 mg l⁻¹ MnCl₂ 4H₂O (Sigma-Aldrich, CAS:13446-34-9), 1 mg l⁻¹ H3BO3 (Sigma-Aldrich, CAS:10043-35-3), 0.4 mg l⁻¹ Na₂MoO₄ 2H₂O (Sigma-Aldrich, CAS:10102-40-6), 3 mg l⁻¹ FeSO₄ 7H₂O (Sigma-Aldrich, CAS:7782-63-0), 0.3 mg l⁻¹ CuSO₄ 5H₂O (Sigma-Aldrich, CAS:7758-99-8), solution adjusted at pH = 4 and stored at 4 °C; 1 mg l⁻¹ Thiamine-HCl (Sigma-Aldrich, CAS:67-03-8); 0.04 g l⁻¹ amino acid mix so that L-Alanine (Sigma-Aldrich, CAS:56-41-7), L-Proline (Sigma-Aldrich, CAS:147-85-3), L-Threonine (Sigma-Aldrich, CAS:72-19-5), Glycine (Sigma-Aldrich, CAS:56-40-6) were each at a final concentration of 5 mg l⁻¹ in the medium. The additional carbon sources that could be added were individually set to a final concentration of 0.4 g l⁻¹. The pH was adjusted at 7.4 prior to a 0.22 μm filter sterilization of the medium. Pre-cultures were recovered from glycerol −80 °C stocks, grew in Luria-Bertani (LB) broth overday for 7 h, then used as 5 μl inoculate in 200 μl M9 (supplemented with variable compounds) in 96 U-bottom wells plates overnight for 14 h. Then 5 μl of each well was passed to a replicate of the plate on the next day for growth monitoring. The temperature was set to 37 °C in a plate reader (Agilent Technologies, BioTek HTX Synergy), with continuous orbital shaking at maximum speed, allowing aerobic growth for 24 h. A monitoring every 10 min of the optical density at 600 nm (OD₆₀₀) was performed. A figure for summarizing the experimental workflow is available in Fig. S11.

## Growth rates determination

The maximal growth rate was determined by sliding a window of 1 h-size, performing a linear regression on the log(OD₆₀₀) data in each window. We then retrieve the maximum specific growth rate as the maximum regression coefficient over all windows. If several growth phases are visible, one can omit a part of the growth curve for the maximal growth rate determination (for this study we always retrieved the maximal growth rate on the first growth phase, so as we are certain that the media contains all added carbon sources). Eight replicates for each medium composition were performed (on a single column of a 96-well plate). Outliers were manually removed after visual inspection of the growth curves or clear statistical deviation of the computed growth rate from the remaining replicates. The numbers of replicates kept range from 2 to 8, with an average of 4.6 (±1.6) replicates per medium composition. Means and standard deviations over replicates were computed to be used for training AMNs and making figures. All raw data and the code to process it are available in the Github repository[43] (cf. "Code availability").

## External training sets acquisition

**Growth rates of *E. coli* metabolic gene KO mutants.** The dataset was downloaded from the ASAP database (Mutant Biolog Data I for K-12

mutant strains)[28]. That dataset was pre-processed by applying several filtering steps: removing substrates that do not appear in iML1515 as possible substrates for uptake fluxes, removing genes not found in iML1515, and removing all data duplicates to obtain a balanced and coherent dataset. The filtered dataset contains 17,400 growth rates: 145 *E. coli* mutants (each having a KO of a single metabolic gene) grown in 120 conditions (each with a different set of substrates) from Biolog phenotype microarrays[44]. The final training set can be found in the source data, provided as a Source Data file (cf. "Data availability").

For practical reasons, we converted the information about metabolic gene KOs information into binary vectors describing which reactions are directly affected by a gene KO, called **R_KO** in Fig. 4. This mapping was automated with iML1515's ability to link genes and reactions. For reactions performed by enzymes encoded by more than one gene, we make the assumption that when any of these genes is knocked-out, the reaction is also knocked-out.

For the FBA computation (Fig. 4c, e), we set an arbitrary upper bound on uptake fluxes (11 mmol gDW⁻¹ h⁻¹ was found to be the best value in terms of regression performance) for each substrate in the dataset when it is present (otherwise 0). To simulate a KO, we set the lower and upper bound of a reaction to zero.

To transform the measured growth rates from continuous values into binary values (for the ROC curves in Fig. 4d, e), following Orth et al.[30], we applied a threshold of 0.165 h⁻¹, which is equal to 5% of the maximum growth rate (3.3 h⁻¹) found in the dataset. Therefore, the classification task can be seen as the ability for the model to classify growth rates below and above the threshold value.

**P. putida growth assays.** The dataset used to generate Fig. 5 (panels e and f) was taken from the study of Nogales et al.[33] presenting iJN1462 (an updated version called iJN1463 is available on BiGG[26]) for *P. putida's* GEM. This state-of-the-art GEM of *P. putida* KT2440 contains a few hundred more genes and reactions from the previous models, allowing better coverage. The dataset corresponds to growth assays with 188 carbon and 108 nitrogen sources. For each condition, we verified that an uptake reaction flux was present in the iJN1463[33] model. Fifty-five conditions contained a nutrient source without a corresponding uptake reaction in the model. For all those conditions, the AMN input would be the minimal medium. In order to avoid biasing the training set with 55 identical conditions, we kept one condition describing the minimal medium for carbon sources and one condition describing the minimal medium for nitrogen sources. The 55 conditions were added back to compute the final score. The training set can be found in the source data, provided as a Source Data file (cf. "Data availability").

The minimal medium assumed in our simulations was taken from Nogales et al.[33], reporting a set of uptake fluxes upper bounds. When testing a carbon (nitrogen) source, glucose (NH₄) was removed from the minimal medium, and the respective nutrient source metabolite was added. Using these simulated growth media, accuracies on growth predictions using Cobrapy (Fig. 5f) were calculated considering as positive all non-zero growth predictions. Results presented in Fig. 5e were obtained by training a reservoir on simulations as explained in Methods "AMN and ANN training parameters". Thus, this reservoir was used to fit experimental data, and **V_in** was directly used as an input for Cobra.

## Statistics and reproducibility

As stated in the previous section "Generation of training sets with FBA", the exchange reactions upper bounds were randomized to produce FBA-simulated training sets. No statistical method was used to pre-determine the sample size, which was chosen based on time and resources. Blinding was not relevant to generate these training sets.

As stated in the previous section "Generation of an experimental training set", the media of the experimental training set were randomized by randomly drawing carbon sources combinations. The growth

rate measures were computed as means over 2 to 8 technical replicates. In all figures displaying this dataset, we show the standard deviation over replicates as the error bars. No statistical method was used to predetermine the sample size of 110. This size was chosen based on time and resources. Blinding was not relevant to generate this training set.

As stated in the previous section "External training sets acquisition", we used two publicly available datasets, for which the authors did not specify any statistical method to predetermine the sample size. To our knowledge, there was no replication scheme for these external training sets. Incompatible data were excluded in the pre-processing steps of the training set stemming from the ASAP database (cf. "External training sets acquisition" section). For more details on the statistics and reproducibility of these external training sets, please refer to the original studies.

### Reporting summary
Further information on research design is available in the Nature Portfolio Reporting Summary linked to this article.

## Data availability
Metabolic models used in this study can be found with the following accessions on the BiGG database[26]: *E. coli* core (http://bigg.ucsd.edu/models/e_coli_core), iML1515, iJN1463. Unidirectional versions of these models can be found on our repository at https://github.com/brsynth/amn_release/tree/main/Dataset_input/. The original dataset from the ASAP database[28] can be found under the accession Mutant Biolog Data I (https://asap.genetics.wisc.edu/asap/experiment_data.php). The original dataset from Nogales et al.[33] can be found in Supporting Information's Table S2 of the study. The source data underlying all figures presented in the main manuscript and Supplementary Information (including training sets used in Figs. 3–5), are provided with this paper as a downloadable archive. Additional datasets and raw data are available on our Github repository (cf. "Code availability"), or from the corresponding authors upon request. Source data are provided with this paper.

## Code availability
All scripts and data for generating results presented in this paper are available within a documented repository. For a citable and stable version of the repository supporting this article, refer to our repository[43] hosted on Zenodo with the https://doi.org/10.5281/zenodo.8056442 (https://zenodo.org/record/8056442). Alternatively, to access future releases and interact with the repository authors, refer to Github (https://github.com/brsynth/amn_release). The repository includes tutorials in Google Colab notebooks. The released codes make use of Cobrapy[19], numpy[45], scipy[46], pandas[47], tensorflow[48], sci-kit learn[27] and keras[42] libraries. Figures were generated using the matplotlib[49] and seaborn[50] libraries.

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

## Acknowledgements

J.L.F. would like to acknowledge funding provided by the ANR funding agency grant numbers ANR-18-CE44-0015 (SynBioDiag project) and ANR-21-CE45-0021-01 (AMN project) and the UE HORIZON BIOS program (grant number 101070281). L.F. is supported by INRAE's MICA department and by INRAE's metaprogram DIGIT-BIO. B.M. is supported by an Ecole Normale Supérieure (ENS) Scholarship. We thank Aymeric Gaudin (CentraleSupélec Engineering School) for early development in reservoir computing with AMN, Ivan Radkevich (University of Paris Saclay) for his work on custom RNN cells and Tom Lorthios and Hadi Jbara (AgroParisTech and University of Paris Saclay) for their help on collecting data for experimental training sets. We thank Anne Giralt and Laetitia Laversa (INRAE) for reading and improving our manuscript.

## Author contributions

L.F. and J.L.F. wrote the core of the text of the manuscript. J.L.F. designed the study and wrote the Wt and QP-solver and all the AMN and AMN-Reservoir codes used to produce results presented in Figs. 1–5. B.M. wrote the LP-solver and the corresponding part in the "Methods" section and Supplementary Information. L.F. benchmarked all codes, wrote the codes transforming SBML models into unidirectional networks and processing experimental data, and handled the Colab and Git implementations. L.F. also performed the experimental work reported in Fig. 3, acquired the data and run the AMN and AMN-Reservoir codes to produce Figs. 4 and 5 and wrote the corresponding "Methods" section. W.L. contributed to designing the project and was involved in the discussions and writing the manuscript. All authors read, edited, and approved the manuscript.

## Competing interests

The authors declare no competing interests.
