## [Peer review file · Nature Communications]

REVIEWER COMMENTS

Reviewer #1 (Remarks to the Author):

This paper introduces new methodology utilizing a hybrid of neural networks and metabolic network models for improved prediction of experiments involving metabolic networks. In particular, there is novelty in the approach of using physics informed neural networks to introduce a method for performing backpropagation to train the hybrid models within the context of constrained optimization. The new method is likely to have broad applicability to other metabolic networks not considered in the paper, as well as gene regulatory network modeling. I have two minor comments/suggestions for the paper:

1) In the results section "AMNs can be used in a reservoir computing framework to enhance the predictive power of traditional FBA solvers" the authors utilize their previously trained AMN network and parameters as a reservoir within a reservoir computing model. It seems like the authors are implying that transfer learning of the learned AMN parameters is crucial for having high predictive accuracy of the AMN-reservoir model. However, the usual method for reservoir computing models is that the reservoir is simply initialized to completely random weights with some constraints on the spectral radius of the matrix encoding the reservoir network. Did the authors try to use random initialized weights instead of the AMN pretrained weights? To answer this question, I suggest at least some minor computational study showing that it is necessary to use AMN pretrained parameters instead of randomly initialized reservoir parameters. Such a study would include various reservoir sizes.

2) Neural networks are typically initialized with random weights and biases. Thus, there could be variability in the results presented in the paper due to neural network initialization. Both the ANN and AMN model predictions might be improved by running the training many times, for example 20 or 30 times, and taking the best performing result. Did the authors perform their study with randomly initializing the ANNs several times, or are the results presented from a single initialization?

Reviewer #2 (Remarks to the Author):

Hybrid models enabling neural computations with metabolic networks
Faure et. al.

The manuscript by Faure and colleagues describes a new approach to modelling metabolic networks combining recurrent neural networks and constraint-based modelling (CBM) into a new hybrid framework called an artificial metabolic network (AMN). The main advantage of this approach is that the input fluxes to CBM can be estimated from data, then FBA applied to predict growth phenotype. The authors compare their new framework to more traditional approaches using simulations and a dataset comprising of E. coli growth rates in different media compositions. Overall, I really like the idea of using RNNs to model the fluxes in metabolic networks. I have some comments below for improvement of the manuscript.

1) I don't believe the title of the manuscript accurately reflects the main points of the paper. The manuscript is not about utilising metabolic networks for computation. I think the authors should rename it to something more appropriate.

2) I think a visualisation of key parts of Table 1 as a bar chart (or similar) is needed as the table right now is very difficult to compare performance of different architectures.

3) In Figure 4 caption, Q2 needs to be redefined. (I presume it is the R2 of the regression).

4) All of the results follow from knowing the exact structure of the metabolic network which is almost impossible (CBM and FBA of course also have this problem). How robust are the results to sampling of the reactions in the models? Is this approach more or less robust than traditional methods?

Reviewer #3 (Remarks to the Author):

This paper proposes a new type of neural network called 'artificial metabolic networks' (ANN) that incorporates a genome scale model into the training of neural networks. I read the paper with great interest as I have experience in both domains, and the idea of integrating FBA-type models directly into neural network training is interesting and novel. The paper, however, is quite unclear in its presentation and demonstrating the utility of the method. Critically, I found the motivation for building such models unclear, particularly because it is unclear what specific problem in biological modelling question/challenge this algorithm intends to solve. The paper is difficult to follow, as in fact four different models are presented and it is difficult to follow the relation between these models and why they are needed paper.

Altogether I don't think this paper is suitable for the journal, as the contribution is rather narrow and of interest to an FBA-specific audience. As I explain below I would advise the authors to focus on the last model 'AMN reservoir' as the main subject, and carry further tests of how this model can outperform classic FBA in a wider range of problems (not just prediction of growth rate from media composition). This would make the results much more relevant to the general metabolic engineering, synthetic biology, and q-bio audiences.

1. The introduction states that AMNs aim to solve some of the caveats of FBA, particularly its inability to convert concentrations of growth media components into the boundary fluxes that go into FBA calculations. This caveat is real, but I also found that for the results in Fig 4, the media composition was featurized as a binary vector, without any information about concentrations. The 4th model 'AMN Reservoir' seems to take care of this, but this model is not presented as the main result of the paper. Altogether the paper gets confusing at several points and seems appropriate for a readership already familiar with the inner details of FBA, not potential users of the algorithm in metabolic modelling/engineering.

2. I strongly suggest to focus on AMN reservoir, and take the focus away from the other three algorithms AMN-Wt, AMN-LP and AMN-QP. AMN-Wt suffers an important problem when including exact bounds (as well explained by the paper) and AMN-LP vs AMN-QP differ their computational implementation (linear vs quadratic programming) with no difference in performance (Fig 4). As it stands this presentation is more suited for a computer science audience.

3. I struggled to find a clear description of what the inputs to the AMNs are. In all examples it seems to be the media composition, but there also extensive references to the boundary fluxes being the inputs. This needs to be clarified, perhaps with a figure to replace Fig 1 (which is technical and does not help to explain the method).

4. The careful calculations of the gradients are quite clear and the attention to rigour is appreciated. Many papers would overlook these mathematical details, but they are important as the authors clearly appreciate. I found, however, that the mathematical details in the main text tended to obscure the aims of the models and what biological problems can be solved, and what people might use this algorithm for.

5. The Q2 accuracies in Fig 4 are not particularly exciting (~ 0.78 for all methods), but I think a much better benchmark would be to compare it to the variability in the experimental repeats as opposed to

a perfect fit $Q^2=1$. In fact, error bars in the growth data are not small which means that $Q^2=0.78$ may actually be much closer to the experimental reproducibility of the measurements. If that's the case, this would be good news as it means the method is in fact more accurate than how it has been presented in the paper.

6. A general comment is that prediction of growth rate from media composition is not the most compelling use case of this method, and I encourage the authors to explore this further. Classic FBA is not that bad at predicting this (or at least the authors should compare AMNs with pure FBA predictions). The latest E coli model from Monk et al already predicts growth rates of WT and knockouts in many growth media with accuracy >90%. The discussion enumerates several problems where the algorithm might be useful and would be good to see at least some of these implemented. It would eg be useful to see how the AMNs fare when using metabolic models that are not as well curated as the E coli one. The latest E coli model from Monk et al has impressive predictive power and would be very interesting if AMNs can improve FBA models for less characterized organisms, as there is where the community really needs a substantial increase in predictive power.

7. More references to recent work on machine learning + FBA were missing, as well as the early attempts to combine the two (eg <https://pubmed.ncbi.nlm.nih.gov/18652654/>). The current literature is quite active as well, eg <https://pubmed.ncbi.nlm.nih.gov/33580712/> and <https://europepmc.org/article/ppr/ppr474123>

8. The introduction of the paper is long and the flow is irregular. Particularly because it includes quite some content about pros/cons of the method that would fit better in the discussion. The intro should just tell us what problem is being solved, how it is currently solved, and how this method offers an improvement on current solutions.

9. To assess the power of AMNs, I advise to also benchmark against classic non-neural machine learning regressors. The comparison with ANNs is not very fair, because it is likely these will overfit the data. For a dataset like in Fig 4 (N=110 measurements with p=10 binary features), it would seem unwise to use an ANN of such dimensions for regression. One alternative is to use random forest or support vector methods as a benchmark. Another alternative is to use grid search or bayesian optimization to determine the ANN hyperparameters and architecture. No details on hyperparameter selection were given, which is very unusual for this type of paper. At the minimum, the paper should explain how the architecture of the ANNs were chosen and show some evidence that they are better than other choices (in terms of Q^2 and overfitting, for example).

Point-by-point response to reviewers

Reviewer #1 (Remarks to the Author):

This paper introduces new methodology utilizing a hybrid of neural networks and metabolic network models for improved prediction of experiments involving metabolic networks. In particular, there is novelty in the approach of using physics informed neural networks to introduce a method for performing backpropagation to train the hybrid models within the context of constrained optimization. The new method is likely to have broad applicability to other metabolic networks not considered in the paper, as well as gene regulatory network modeling. I have two minor comments/suggestions for the paper:

Response: We thank the Reviewer for the positive feedback. We indeed do hope that our study will inspire others to use similar ideas with different metabolic or gene regulatory networks. Also, the Reviewer's comments encouraged us to push further some analyses on our models, and we hope the new results show convincingly that our reservoir framework and network initializations are robust.

1) In the results section "AMNs can be used in a reservoir computing framework to enhance the predictive power of traditional FBA solvers" the authors utilize their previously trained AMN network and parameters as a reservoir within a reservoir computing model. It seems like the authors are implying that transfer learning of the learned AMN parameters is crucial for having high predictive accuracy of the AMN-reservoir model. However, the usual method for reservoir computing models is that the reservoir is simply initialized to completely random weights with some constraints on the spectral radius of the matrix encoding the reservoir network. Did the authors try to use random initialized weights instead of the AMN pretrained weights? To answer this question, I suggest at least some minor computational study showing that it is necessary to use AMN pretrained parameters instead of randomly initialized reservoir parameters. Such a study would include various reservoir sizes.

Response: We thank the Reviewer for this interesting remark on our use of reservoirs. First, we would like to clarify that we used the term 'reservoir' in 'AMN-Reservoir' to emphasize the fact that parameters are frozen. Yet, as rightly indicated by the Reviewer, the parameters of this part of the model are not randomly chosen, but are instead learned from reference FBA simulated data during a pre-training step. To directly respond to the Reviewer's question, we did a new test in which we randomly initialized our AMN-Reservoirs with weights drawn from a normal probability law $N(0, 0.05)$. Then, we used our experimental dataset (the 110 growth rates measured in different media), to train a prior layer, as in our regular AMN-Reservoir results (shown in Figure 5), to fit and predict the growth rate. This yielded very poor performance for both fitting (R^2) and predicting (Q^2), even with different network parameters (number of layers, layer sizes). Results can be found in the Table below.

Hidden layer number	Hidden layer size	R ²	Q ²
1	50	-2.4 (+/- 0.6)	-3.0 (+/- 1.2)
	100	-2.0 (+/- 1.1)	-2.7 (+/- 1.6)
	500	-2.1 (+/- 1.4)	-2.9 (+/- 2.5)
5	50	-7.6 (+/- 0.5)	-8.3 (+/- 1.9)
	100	-6.8 (+/- 1.2)	-7.6 (+/- 2.2)
	500	-7.1 (+/- 1.1)	-7.9 (+/- 2.3)

Overall, the above Table shows that training an AMN-Reservoir on FBA simulations is crucial to the final performance of the model, and we actually need this pre-training not only to extract the best inputs for a CBM framework (V_{in}) but also for good predictive accuracy.

Manuscript Modifications: We added the following sentences on page 15: “Loosely inspired by reservoir computing, we call this architecture ‘AMN-Reservoir’ (Figures 1d and 5a). Let us note nonetheless that we are not using usual reservoirs with random weights and a post-processing trainable layer. As a matter of fact, we do not reach satisfactory performances when we substitute the AMN-Reservoir weights (learned during training) by random weights.”

2) Neural networks are typically initialized with random weights and biases. Thus, there could be variability in the results presented in the paper due to neural network initialization. Both the ANN and AMN model predictions might be improved by running the training many times, for example 20 or 30 times, and taking the best performing result. Did the authors perform their study with randomly initializing the ANNs several times, or are the results presented from a single initialization?

Response: We wish to clarify that for most results presented in the manuscript, we repeated cross-validation several times. For each of these repeats, the random seed was changed, which impacted the network initializations (weights and biases) and training set vs. validation set contents. For instance, over 3 repeats of 10-fold cross-validations, we generated altogether 30 different training sets, 30 different validation sets, and 30 different models with different initializations. Below is a Table showing a statistics summary for the Q² of 20 cross-validation repeats with the data and model used to produce Figure 3a. We note that the best model (max) yields performances within those indicated on Figure 3a (Q² = 0.78 ± 0.03) and carried out on 3 repeats only.

	Q ²
Mean	0.77
std	0.017
min	0.73
25%	0.76

50%	0.77
75%	0.77
max	0.79

Manuscript Modifications: We added on page 11 the following sentences: *"For displaying meaningful results, and to avoid any overfitting bias, we show in Figure 3 predictions for points unseen during training. More precisely, we computed the mean and standard deviation of predictions over 3 repeats of stratified 10-fold cross-validations, each repeat having all points predicted, by aggregating validation sets predictions of each fold. Overall, results presented in Figure 3 have been compiled over $3 \times 10 = 30$ different AMN models, each having different random seeds for the neural layer initialization and the train/validation splits"*.

Reviewer #2 (Remarks to the Author):

The manuscript by Faure and colleagues describes a new approach to modelling metabolic networks combining recurrent neural networks and constraint-based modelling (CBM) into a new hybrid framework called an artificial metabolic network (AMN). The main advantage of this approach is that the input fluxes to CBM can be estimated from data, then FBA applied to predict growth phenotype. The authors compare their new framework to more traditional approaches using simulations and a dataset comprising of E. coli growth rates in different media compositions. Overall, I really like the idea of using RNNs to model the fluxes in metabolic networks. I have some comments below for improvement of the manuscript.

Response: We would like to thank the reviewer for the interest and appreciation of our manuscript, as well as for the very helpful suggestions to improve readability and a clear presentation of our results.

1) I don't believe the title of the manuscript accurately reflects the main points of the paper. The manuscript is not about utilizing metabolic networks for computation. I think the authors should rename it to something more appropriate.

Response: We changed the title, which is now focused on the motivation of our work.

Manuscript Modifications: new title 'A neural-mechanistic hybrid approach improving the predictive power of genome-scale metabolic models'

2) I think a visualisation of key parts of Table 1 as a bar chart (or similar) is needed as the table right now is very difficult to compare performance of different architectures.

Response: We agree and replaced Table 1 (now Supplementary Table S1) with a Figure (new Figure 2). We show in this Figure loss and Q^2 values across learning epochs. Thus, the reader can compare performances for different inputs (UB, EB, E. coli core/iML1515) and different models (AMN-QP/LP/Wt, and ANN). We also added (to now Table S1) information on network architectures and hyperparameters (as requested by Reviewer 3).

Manuscript Modifications: We added a new Figure 2 and Table 1 was moved to Supplementary Information (now Table S1).

3) In Figure 4 caption, Q2 needs to be redefined. (I presume it is the R2 of the regression).

Response: Indeed, Q_2 stands for R^2 computed on predictions only. This was mentioned in the abbreviation section, but for clarity we now added a definition where the symbol is first used.

Manuscript Modifications: A description of Q_2 was added to Figures 2-4 captions.

4) All of the results follow from knowing the exact structure of the metabolic network which is almost impossible (CBM and FBA of course also have this problem). How robust are the results to sampling of the reactions in the models? Is this approach more or less robust than traditional methods?

Response: We fully acknowledge that the metabolic model reconstructions we use for our hybrid models may be imperfect. However, this is a general problem that does not only concern our hybrid models, and our modeling approach is not designed to remedy faulty network structures, at least not in the current study. We therefore need to trust metabolic models in the context of hybrid modeling.

Following the reviewer's comment, we brainstormed how, in principle, the effects of network errors within our approach could be assessed. In theory, we could randomly subsample reactions of a metabolic network, obtaining sub-networks of different sizes, and test the performances of hybrid models on these. However, the results would be hardly interpretable since metabolic network reconstructions rely on long manual curation and random subsampled models would strongly differ from such hand-curated models in their biological properties. Another, more reasonable test would be to compare (curated) networks of different size and to compare the results. As a matter of fact, we present in the study results with metabolic networks of different sizes: *E. coli* core having 154 reactions, and iML1515 having 3682 reactions; both giving satisfactory performances. Thus, we can assume our method is reasonably robust against adding or removing peripheral regions of a network.

But robustness of our method is not an end in itself. In fact, certain changes (such as individual knockouts) should have an impact on our results. A result we added new Figure 4 (responding to Reviewer 3) showing that classical CBM/FBA is very sensitive to reaction KOs. Indeed, CBM/FBA often leads to infeasible metabolic states that experimentally are feasible (*i.e.*, the organism can actually grow but CBM/FBA predicts no growth). With AMNs, we treat the constraints in a more "tolerant" way: the solution search space is not strictly constrained, but instead the optimization looks for the best tradeoff on the solution fluxes, between fitting experimental data and respecting the metabolic network constraints. As a result, we can avoid these infeasible states and enhance the prediction performance. The results shown in new Figure 4 demonstrate our AMNs are more robust to reaction KOs than classical FBA when considering the experimental data as the ground truth to be respected (AUC=0.9 for AMN vs. AUC=0.71 for FBA).

Coming back to point 4, it is interesting to note that we might be able to develop a method to enhance metabolic network reconstruction based on quantitative prediction performances. Indeed, the hybrid models we developed could be used to screen putative reactions in a metabolic model so that it best matches experimental data.

Manuscript Modifications: We added new Figure 4 and the following sentences in the Discussion section page 19: *"Another possible application of AMN hybrid models is to enhance GEMs reconstruction based on quantitative prediction performance. Indeed, the method we developed for KOs could be adapted to screen putative reactions in a metabolic model so that its predictions match experimental data. This task should be performed after a manual curation, of course, to rely on existing literature knowledge and databases"*.

Reviewer #3 (Remarks to the Author):

This paper proposes a new type of neural network called 'artificial metabolic networks' (ANN) that incorporates a genome scale model into the training of neural networks. I read the paper with great interest as I have experience in both domains, and the idea of integrating FBA-type models directly into neural network training is interesting and novel. The paper, however, is quite unclear in its presentation and demonstrating the utility of the method. Critically, I found the motivation for building such models unclear, particularly because it is unclear what specific problem in biological modelling question/challenge this algorithm intends to solve. The paper is difficult to follow, as in fact four different models are presented and it is difficult to follow the relation between these models and why they are needed paper.

Altogether I don't think this paper is suitable for the journal, as the contribution is rather narrow and of interest to an FBA-specific audience. As I explain below I would advise the authors to focus on the last model 'AMN reservoir' as the main subject, and carry further tests of how this model can outperform classic FBA in a wider range of problems (not just prediction of growth rate from media composition). This would make the results much more relevant to the general metabolic engineering, synthetic biology, and q-bio audiences.

Response: We thank the reviewer for these helpful comments. In our revised manuscript, we first state in the introduction our motivation, focusing on biological questions, and clarify our methods and the relationship between the models. Later we extend the usage of our AMN and AMN-Reservoir beyond growth rate predictions adding the prediction of other fluxes and the predictions of phenotypes with gene KO strains. We also apply our AMN-Reservoir method with a new *P. putida* experimental dataset. Altogether, compared to the previous manuscript version we added three additional experimental datasets on which our AMN and AMN-reservoir are tested. Importantly, in addition to new results, we re-organized the introduction, the Figures and the Discussion, and we moved technical details from the Methods to Supplementary Information to make the paper more accessible and convincing to a large and diverse scientific audience.

Summary of main modifications:

- The title was changed and the introduction was re-written to focus on motivation and biological questions, text was moved from introduction to the discussion.
- A new Figure 1 was added (old one can be found the Supplementary Information, Figure S1).
- A new Figure 2 was added to replace Table 1 now in Supplementary Information (a modified version of old Figure 2 can be found in Supplementary Information Figure S6).
- A new Figure 4 was added presenting results for gene knock-outs experimental data.
- Figure 5 was completed with new results for *P. putida*.
- Mathematical details have been moved from Methods to Supplementary Information.

- A new Supplementary Figure (Figure S7) presenting results of a grid search for architectures and hyperparameters.
- A new Supplementary Figure (Figure S8) was added presenting AMN performance on an experimental multiple fluxes dataset.

1. The introduction states that AMNs aim to solve some of the caveats of FBA, particularly its inability to convert concentrations of growth media components into the boundary fluxes that go into FBA calculations. This caveat is real, but I also found that for the results in Fig 4, the media composition was featurized as a binary vector, without any information about concentrations. The 4th model 'AMN Reservoir' seems to take care of this, but this model is not presented as the main result of the paper. Altogether the paper gets confusing at several points and seems appropriate for a readership already familiar with the inner details of FBA, not potential users of the algorithm in metabolic modelling/engineering.

Response: The Reviewer rightly pointed out that in the initial version of our manuscript, inputs were presented in a confusing manner. We now clarified this, in particular by adding new Figure 1 detailing the inputs for all methods used in the manuscript.

In Figure 1 (panel a), the inputs for the FBA classical method is the vector V_{in} of medium uptake fluxes bounds.

In Figure 1 (panel b), the inputs of the three mechanistic models surrogating FBA is a vector of initial fluxes values (V^0) respecting the boundary constraints (V_{in}).

In Figure 1 (panel c), there are two types of inputs for AMNs: (i) exact or upper bounds on uptake fluxes when the training set is composed of FBA simulated data (V_{in}) or (ii) experimental medium composition for experimental training sets (C_{med}).

In Figure 1 (panel d), the inputs for the 'AMN-Reservoir' are the media compositions (C_{med}). The AMN-Reservoir method follows a 2-step learning process: first an ANN is trained on FBA simulations then on experimental media composition after fixing the parameter of the ANN. After this 2-step learning we extract V_{in} inputs and used it with classical-FBA. The AMN-Reservoir performances (Figure 5 and Supplementary Figure S9) are similar to those of Figure 3 where we directly trained the AMN on experimental conditions. However, direct AMN training does not allow us to extract the V_{in} bound values.

The medium composition (C_{med}) in the new Figure 3 had been labeled 'concentrations' in the initial version of the manuscript, but as noted by the Reviewer, we used in fact a binary vector only. Considering that in our study, all concentrations are equal for all compounds added in the media, there is no need to consider actual concentration values as these turned out to be scaled by the neural layer. Consequently, in the current manuscript version, we use the term "composition" instead of "concentrations".

Manuscript Modifications: We added new Figure 1 along with a new section 'Overview of AMN hybrid models' on pages 5 and 6 explaining our methods and their inputs and we change 'concentration' by 'composition' in the whole manuscript and Figures 1, 3, 4, 5.

2. I strongly suggest to focus on AMN reservoir, and take the focus away from the other three algorithms AMN-Wt, AMN-LP and AMN-QP. AMN-Wt suffers an important problem when including exact bounds

(as well explained by the paper) and AMN-LP vs AMN-QP differ their computational implementation (linear vs quadratic programming) with no difference in performance (Fig 4). As it stands this presentation is more suited for a computer science audience.

Response: The text and Figures presented in the revised manuscript are now focused on the handling of experimental datasets by AMNs and AMN-Reservoir rather than the algorithms of the Wt, LP and QP solvers. To that end, and also answering point 6 below, we added 3 new datasets used to train AMN and AMN-Reservoir. While details and results regarding the Wt, LP and QP solvers are no longer shown in the main manuscript, these methods are still necessary to build AMNs and AMN-Reservoirs and have been moved to Supplementary Information. Answering point 4 below, we also moved all technical details and mathematical equations on the 3 solvers to Supplementary Information.

Manuscript Modifications:

- Old Figure 1 is moved in Supplementary Information (Figure S1) along with a detailed explanation on the functioning of this solver (pages 2, 3 in Supplementary Information)
- Old Figure 2 is moved in Supplementary Information (Figure S6) along with a detailed explanation in section 'MM solver benchmarking'.
- We added to Figure 3 some succinct mathematical details as cartoons for each method.
- Most of the mathematical expressions regarding Wt, LP and QP solvers have been removed from the main manuscript and can now be found in the Supplementary Information (in sections Wt-solver equations, LP-solver equations and QP-solver equations).
- Answering point 6, we added new Figure 4 and complemented Figure 5 where AMN and AMN-Reservoirs are used with new experimental datasets.

3. I struggled to find a clear description of what the inputs to the AMNs are. In all examples it seems to be the media composition, but there also extensive references to the boundary fluxes being the inputs. This needs to be clarified, perhaps with a figure to replace Figure 1 (which is technical and does not help to explain the method).

Response: As already mentioned in our answer to point 1, we replaced our old Figure 1 (now Supplementary Figure S1) with a new one thoroughly describing the inputs. Importantly, AMNs are not generally restricted to a single input data type used, unlike FBA, and this is one strength of the method. Indeed, one could think of other physical parameters such as light exposure, temperature, shaking speed etc., as possible inputs for an AMN. In our study, we used exact or upper bounds on uptake fluxes as inputs when learning on simulated training sets (Figure 2 and Supplementary Table S1), but also medium composition as inputs when learning on experimental data (Figure 3 and Figure 5) and a combination of medium composition and the list of gene knockouts (new Figure 4 and Supplementary Figure S8)

Manuscript Modifications: we added new Figure 1 along with a new section 'Overview of AMN hybrid models' on pages 5 and 6 explaining our methods and their inputs (cf. point 1) and we modified all Figures to clearly identify the inputs of the models.

4. The careful calculations of the gradients are quite clear and the attention to rigour is appreciated. Many papers would overlook these mathematical details, but they are important as the authors clearly appreciate. I found, however, that the mathematical details in the main text tended to obscure the aims of the models and what biological problems can be solved, and what people might use this algorithm for.

Response: The mathematical details have now been moved in Supplementary Information.

Manuscript Modifications: As already mentioned with point 2, most of the mathematical expressions in Methods can now be found in the Supplementary Information (in sections Wt-solver equations, LP-solver equations and QP-solver equations).

5. The Q2 accuracies in Figure 4 are not particularly exciting (~0.78 for all methods), but I think a much better benchmark would be to compare it to the variability in the experimental repeats as opposed to a perfect fit $Q^2=1$. In fact, error bars in the growth data are not small which means that $Q^2=0.78$ may actually be much closer to the experimental reproducibility of the measurements. If that's the case, this would be good news as it means the method is in fact more accurate than how it has been presented in the paper.

Response: We thank the reviewer for this very helpful suggestion about our performance metric. Following the reviewer's recommendation, to account for inter-replicate experimental variability we propose an approach that we briefly describe in the manuscript and below. For each experimental data point, we randomly draw a new point from a normal distribution with a mean and variance equal to what was experimentally determined for the original point. This point can be considered as a 'resampled' experimental data point. When we do this for all points and compute the Q^2 , repeating this process 1000 times, we obtain a mean $Q^2=0.91$ with a standard deviation of 0.02. Consequently, the best 'experimental variability accounting' Q^2 is 0.91, and our performance of $Q^2=0.77$ or 0.78 must be compared to this value. In addition, substituting each point by boxes of sizes equal to the variances for both measurement and prediction, we find that 79% for AMN-QP (76% for AMN-LP and 74% for AMN-Wt) of the boxes intersect the identity line indicating that these points are correctly predicted within the variances.

Manuscript Modifications: In the text presenting Figure 3 on pages 12 and 13 we described the above approach taking into account experimental variability in our regression performance metrics.

6. A general comment is that prediction of growth rate from media composition is not the most compelling use case of this method, and I encourage the authors to explore this further. Classic FBA is not that bad at predicting this (or at least the authors should compare AMNs with pure FBA predictions). The latest E coli model from Monk et al already predicts growth rates of WT and knockouts in many growth media with accuracy >90%. The discussion enumerates several problems where the algorithm might be useful and would be good to see at least some of these implemented. It would eg be useful to see how the AMNs fare when using metabolic models that are not as well curated as the E coli one. The latest E coli model from Monk et al has impressive predictive power and would be very interesting if

AMNs can improve FBA models for less characterized organisms, as there is where the community really needs a substantial increase in predictive power.

Response: We added new results comparing AMNs and FBA for several estimation tasks that are based on experimental datasets: (i) predicting the effect of gene knocked-out *E. coli* mutants in terms of growth rate and other measured fluxes for two different experimental sets (Figure 4 and Supplementary Figure S8), and (ii) predicting the growth rate of *P. putida* grown on different media (modified Figure 5). *P. putida* was chosen because its iJN1463 model is not as refined as those produced for *E. coli* and because experimental data were available. In both cases, we formulated the problem as a regression task to assess the ability of our hybrid models to predict growth rate and other flux values, and alternatively as a classification task to determine if our hybrid models could accurately predict growth vs. no growth phenotypes.

In all these regression and classification tasks we found that AMNs and AMN-Reservoirs outperform FBA. We also observed that, as mentioned in the Reviewer comment '*Monk et al. already predicts growth rates of WT and knockouts in many growth media with accuracy >90%.*', FBA alone can reach good performance when considering qualitative performance only. Indeed, we observe a 0.71 AUC with FBA for gene knocked-out *E. coli* mutants and 83% accuracy for *P. putida* growth assay (the AMN and AMN-Reservoir performances are respectively for the two datasets 0.90 AUC and 96% accuracy).

Finally, our new Supplementary Figure S8 show AMN performances predicting 31 experimentally measured fluxes, with a regression coefficient on validation sets equals to 0.9 (variance weighted average Q2). These results are further discussed in Supplementary Information section 'AMNs benchmarking with gene knockouts and multiple measured fluxes'. We would also like to point out that in our original manuscript (and in the current version), our AMN-Reservoirs allowed us to compute media uptake fluxes that, when plugged into classical FBA, improve the performance of FBA, showing that AMNs can indeed accurately predict fluxes other than growth rate.

Manuscript Modifications: We added new figures (Figure 4 and Supplementary Figure S8) and complemented Figure 5 with results for *P. putida*. Classical FBA computations (Figure 4c, Figure 4e, Figure 5d, Figure 5f) are compared with the AMN performance (Figure 4b, Figure 4d, Figure 5c, Figure 5e). These new results are discussed on pages 13-17. We also added the section 'AMNs benchmarking with gene knockouts and multiple measured fluxes' in Supplementary Information.

7. More references to recent work on machine learning + FBA were missing, as well as the early attempts to combine the two (eg <https://pubmed.ncbi.nlm.nih.gov/18652654/>). The current literature is quite active as well, eg <https://pubmed.ncbi.nlm.nih.gov/33580712/> and <https://europepmc.org/article/ppr/ppr474123>

Response: We added the above references and others within a new section on the integration of ML with FBA models.

Manuscript Modifications: In the introduction, the following text was added on page 3: "To improve FBA phenotype predictions, ML approaches have been used to couple experimental data with FBA."

Among published approaches, one can cite Plaimas et al.⁶ where ML is used after FBA as a post-process to classify enzyme essentiality. Similarly, Schinn et al.⁷ used ML as a post-process to predict amino-acid concentrations. Freischem et al.⁸ computed a mass flow graph running FBA on the E. coli model iML1515⁹ and used it with a training set of measured growth rates on E. coli gene KO mutants. Several ML methods were then utilized in a post-process to classify genes as essential vs. non-essential. As reviewed by Sahu et al.¹⁰, ML has also been used to preprocess data and extract features prior to running FBA. For instance, data obtained from several omics methods can be fed to FBA, after processing multi-omics data via ML¹¹⁻¹³. In all these previous studies, and as discussed in Sahu et al.¹⁰, the interplay between FBA and ML still shows a gap: some approaches use ML results as input for FBA, others use FBA results as input for ML, but none of them embed FBA into ML, as we do in this study with the Artificial Metabolic Network (AMN) hybrid models. “

8. The introduction of the paper is long and the flow is irregular. Particularly because it includes quite some content about pros/cons of the method that would fit better in the discussion. The intro should just tell us what problem is being solved, how it is currently solved, and how this method offers an improvement on current solutions.

Response: In the revised manuscript, we made the introduction section more concise. We now begin our introduction by outlining the motivation of the work presented. We then state the problem from a machine learning perspective, a constraint-based modeling perspective, and a hybrid model perspective (providing references to previous work) . We end the introduction stating how our hybrid method offers an improvement on current solutions.

We removed from the introduction the section on pros & cons of the constraint-based FBA-like methods, which can now be found in the discussion.

Manuscript Modifications: Our introduction has been re-written and pros & cons of constraint-based FBA-like methods have been moved to the discussion.

9. To assess the power of AMNs, I advise to also benchmark against classic non-neural machine learning regressors. The comparison with ANNs is not very fair, because it is likely these will overfit the data. For a dataset like in Figure 4 (N=110 measurements with p=10 binary features), it would seem unwise to use an ANN of such dimensions for regression. One alternative is to use random forest or support vector methods as a benchmark. Another alternative is to use grid search or bayesian optimization to determine the ANN hyperparameters and architecture. No details on hyperparameter selection were given, which is very unusual for this type of paper. At the minimum, the paper should explain how the architecture of the ANNs were chosen and show some evidence that they are better than other choices (in terms of Q2 and overfitting, for example).

Response: Regarding the use of ANN (simple dense neural networks in our study), we would like to point out that ANNs were not trained on experimental datasets but with large simulation datasets (154000 and 550000 datapoints in new Figure 2 and Supplementary Table S1) to avoid overfitting.

Following the Reviewer suggestion, we now tested traditional ML methods with our smaller experimental datasets. Precisely, using the 110 datapoints of Figure 3, we find with random forest a

performance value $Q^2 = 0.71$ and for Figure 4, we find with XGBoost $Q^2 = 0.75$, in both cases using the same validation sets as for our AMNs. These values are lower than those returned by AMNs, but more importantly, the above traditional ML techniques do not handle mechanistic constraints and consequently predicted flux values that cannot be plugged into classical FBA as we do with the AMN-Reservoir.

As shown in Supplementary Information section ‘AMNs benchmarking varying hyperparameters’ and Figure S7, we used a grid search to find the best architectures and hyperparameters. Our AMN architectures along with the hyperparameters are now listed in Supplementary Table S1 for all FBA-simulated training sets and in the captions of Figs 3-5 for experimental training sets.

Finally, regarding overfitting, we recall that Q^2 is a regression coefficient computed on a cross-validation set or an independent test set (therefore excluding training data), and a Q^2 value is usually low when the training data are overfitted (high R^2 low Q^2).

Manuscript Modifications: We added Supplementary Figure S7 showing hyperparameters grid search and added architecture and hyperparameters in Table S1 and in the captions of Figures 3-5. We also added in the text presenting Figures 3 and 4 the regression performance of respectively Random Forests and XGBoost for the same datasets and cross-validation schemes:

Page 12: “As a matter of comparison, a decision tree algorithm predicting only the growth rate with C_{med} (the ‘RandomForestRegressor’ function from the sci-kit learn package²⁷ having 1,000 estimators and other parameters left with default values) reach a regression performance of 0.71 ± 0.01 with the same dataset and cross-validation scheme, indicating AMNs can outperform regular machine learning algorithms.”

Page 14: “The AMN regression performance in Figure 4 (aggregated predictions from a 10-fold cross-validation) reaches $Q^2=0.81$ (Figure 4b). For comparison, a decision tree algorithm predicting only the growth rate from C_{med} and R_{KO} (the ‘XGBRegressor’ function from the XGBoost package²⁹ with all parameters set to default values) yields a regression performance of 0.75, with the same cross-validation scheme and dataset.”

REVIEWERS' COMMENTS

Reviewer #1 (Remarks to the Author):

The authors have satisfactorily addressed my previous concerns. I recommend accepting the paper for publication.

Reviewer #2 (Remarks to the Author):

My suggestions have all been adequately addressed. I believe the paper is much improved due to the comments of all reviewers.

Reviewer #3 (Remarks to the Author):

This manuscript is much improved now and the authors have taken on board most of our feedback, which is great. The contents and message are more streamlined now and I do not have any further comments on the main text. Regarding the abstract, I suggest to simplify the text and avoid listing too many performance metrics.

Point-by-point response to reviewers

Reviewer #1 (Remarks to the Author):

The authors have satisfactorily addressed my previous concerns. I recommend accepting the paper for publication.

We thank the Reviewer for this positive feedback.

Reviewer #2 (Remarks to the Author):

My suggestions have all been adequately addressed. I believe the paper is much improved due to the comments of all reviewers.

We would like to thank the Reviewer and fully agree the paper has been improved taking into account the reviewers' comments.

Reviewer #3 (Remarks to the Author):

This manuscript is much improved now and the authors have taken on board most of our feedback, which is great. The contents and message are more streamlined now and I do not have any further comments on the main text. Regarding the abstract, I suggest to simplify the text and avoid listing too many performance metrics.

We thank the Reviewer for these comments. We have removed performance metrics from the abstract, and per editorial request, shorten the abstract to less than 150 words.